EMBO
Molecular Medicine

# FOXF1 promotes tumor vessel normalization and prevents lung cancer progression through FZD4

Fenghua Bian [1,9], Chinmayee Goda [1,9], Guolun Wang[1,9], Ying-Wei Lan[1,2], Zicheng Deng[1,2], Wen Gao [2], Anusha Acharya[1], Abid A Reza [1], Jose Gomez-Arroyo[1], Nawal Merjaneh[3], Xiaomeng Ren [4], Jermaine Goveia[5], Peter Carmeliet [5,6], Vladimir V Kalinichenko[2,7] & Tanya V Kalin [1,2,3,8 ✉]

## Abstract

Cancer cells re-program normal lung endothelial cells (EC) into tumor-associated endothelial cells (TEC) that form leaky vessels supporting carcinogenesis. Transcriptional regulators that control the reprogramming of EC into TEC are poorly understood. We identified Forkhead box F1 (FOXF1) as a critical regulator of EC-to-TEC transition. FOXF1 was highly expressed in normal lung vasculature but was decreased in TEC within non-small cell lung cancers (NSCLC). Low *FOXF1* correlated with poor overall survival of NSCLC patients. In mice, endothelial-specific deletion of FOXF1 decreased pericyte coverage, increased vessel permeability and hypoxia, and promoted lung tumor growth and metastasis. Endothelial-specific overexpression of FOXF1 normalized tumor vessels and inhibited the progression of lung cancer. FOXF1 deficiency decreased Wnt/β-catenin signaling in TECs through direct transcriptional activation of *Fzd4*. Restoring FZD4 expression in FOXF1-deficient TECs through endothelial-specific nanoparticle delivery of *Fzd4* cDNA rescued Wnt/β-catenin signaling in TECs, normalized tumor vessels and inhibited the progression of lung cancer. Altogether, FOXF1 increases tumor vessel stability, and inhibits lung cancer progression by stimulating FZD4/Wnt/β-catenin signaling in TECs. Nanoparticle delivery of FZD4 cDNA has promise for future therapies in NSCLC.

**Keywords** Tumor-Associated Endothelial Cells; Wnt Signaling; Fzd4; Foxf1; Nanoparticle Delivery System
**Subject Categories** Cancer; Respiratory System

## Introduction

Lung cancer is the leading cause of cancer-related mortality worldwide. Current treatment strategies include chemotherapy and/or radiotherapy, and surgery in the case of patients diagnosed with early-stage lung cancer. However, the 5-year survival rate of patients with advanced NSCLC remains less than 20% (Siegel et al, 2019), emphasizing a need to develop better treatment strategies. Interactions between the tumor microenvironment and tumor cells play a crucial role in tumor progression. Tumor cells secrete pro-angiogenic factors, like VEGF, that recruit endothelial cells to form tumor-associated vasculature, which provides nutrients and oxygen to the proliferating tumor cells, thereby supporting tumor growth. However, antiangiogenic therapies have failed to improve overall survival of lung cancer patients (Alshangiti et al, 2018), suggesting that a deeper understanding of tumor-associated vascular biology is required.

Tumor-associated blood vessels are structurally and functionally abnormal. The presence of tumor cells induces reprogramming of normal endothelial cells (EC) into tumor-associated endothelial cells (TEC) (Dudley, 2012). While ECs are quiescent and form tight endothelial barrier, TECs are activated and form leaky vessels that support carcinogenesis. Re-programmed TEC secrete pro-tumorigenic mediators, supporting tumor cell proliferation and invasion (Maishi and Hida, 2017; Zhao et al, 2016). TEC are resting on a porous basement membrane with decreased pericyte coverage, causing increased vessel permeability. Increased vessel permeability induces tumor-associated inflammation and contributes to increased intravasation and extravasation of tumor cells, stimulating tumor invasion and metastasis (Kienast et al, 2010). Perfusion through tumor-associated vessels is decreased, causing hypoxia within the tumors (Jain, 2014). The hypoxic and acidic tumor microenvironment supports an immunosuppressive environment and prevents the activity of cytotoxic T cells, thereby promoting tumor growth (Huang et al, 2018).

Normalization of tumor vessels' structure and function alleviates hypoxia in the tumor microenvironment and increases the efficacy

[1]Division of Pulmonary Biology, Cincinnati Children's Hospital Medical Center, 3333 Burnet Ave., Cincinnati, OH 45229, USA. [2]Department of Child Health, Phoenix Children's Research Institute, University of Arizona College of Medicine-Phoenix, 475 N 5th Street, Phoenix, AZ 85004, USA. [3]Center for Cancer and Blood Disorders, Phoenix Children's Hospital, 1919 E Thomas Rd., Phoenix, AZ 85016, USA. [4]Division of Asthma Research of Cincinnati Children's Hospital Medical Center, 3333 Burnet Ave., Cincinnati, OH 45229, USA. [5]Laboratory of Angiogenesis and Vascular Metabolism, Department of Oncology and Leuven Cancer Institute (LKI), KU Leuven, VIB Center for Cancer Biology, Leuven 3000, Belgium. [6]Center for Biotechnology, Khalifa University of Science and Technology, Abu Dhabi, UAE. [7]Division of Neonatology, Phoenix Children's Hospital, 1919 E Thomas Rd., Phoenix, AZ 85016, USA. [8]Department of Internal Medicine, Division of Pulmonary and Critical Care, University of Arizona College of Medicine-Phoenix, 475 N 5th Street, Phoenix, AZ 85004, USA. [9]These authors contributed equally: Fenghua Bian, Chinmayee Goda, Guolun Wang. ✉E-mail: tatianakalin@arizona.edu

of anti-cancer therapy (Martin et al, 2019). Vascular normalization decreases endothelial permeability, improves the perfusion of tumor-associated blood vessels, and creates an immune-permissive microenvironment that inhibits cancer progression (Hatfield et al, 2015; Lanitis et al, 2015). However, molecular mechanisms critical for tumor vascular normalization are not well understood. In this study, we compared normal EC with TEC from mouse and human lung tumors to identify FOXF1 as a critical transcriptional regulator of EC-to-TEC reprogramming. FOXF1 is an embryonic transcription factor from the Forkhead Box (Fox) family of transcription factors (Clark et al, 1993; Kaestner et al, 1993). *Foxf1*$^{-/-}$ mice are embryonic lethal (Kalinichenko et al, 2001a; Mahlapuu et al, 2001). Deletions or point mutations in *FOXF1* gene locus are linked to Alveolar Capillary Dysplasia with Misalignment of Pulmonary Veins (ACDMPV), a lethal congenital disorder of newborns and infants (Dharmadhikari et al, 2015). Conditional deletion of *Foxf1* from EC caused embryonic lethality due to the inability of FOXF1-deficient EC undergo angiogenesis in response to VEGF signaling (Ren et al, 2014). The role of FOXF1 in lung cancer is unknown.

In this study, we show that FOXF1 is expressed in normal lung EC but is decreased in the tumor-associated vasculature of human and mouse NSCLC. Decreased expression of *FOXF1* correlates with poor prognosis in NSCLC patients. Using transgenic mice in which *Foxf1* was deleted or overexpressed in endothelial cells, we found that FOXF1 inhibits lung tumor growth and metastasis and normalizes tumor-associated blood vessels by maintaining Wnt/β-catenin signaling in ECs through transcriptional activation of FZD4. Our results support the use of FOXF1-activating therapies for vascular normalization in lung cancers.

# Results

## FOXF1 is decreased in human and mouse NSCLC-associated endothelial cells

Endothelial-specific expression of FOXF1 in the lung tissue of NSCLC patients was visualized by co-staining tissue sections for FOXF1 and CD31. In human donor lungs without tumors, FOXF1 protein was detected in a majority of endothelial cells (Fig. 1A, left panels and Fig. 1B). In contrast, only 5–10% of endothelial cells expressed FOXF1 in human lung adenocarcinomas and squamous cell carcinomas (SCC) (Fig. 1A, right panels and Fig. 1B). FOXF1 was not detected in tumor cells (Fig. 1A). FACS-sorted endothelial cells from human NSCLC tumors (TEC) showed a significant decrease in *FOXF1* mRNA compared to endothelial cells isolated from donor's lungs (EC) (Fig. 1C). The Cancer Genome Atlas (TCGA) data mining showed that NSCLC patients ($N = 1925$) with higher *FOXF1* mRNA expression had increased overall survival compared to patients with lower *FOXF1* levels (Figs. 1D and EV1A), and that the survivors had significantly higher levels of FOXF1 compared to non-survivors (Fig. 1E). Thus, FOXF1 is decreased in NSCLC-associated endothelial cells, and its expression levels correlate with poor prognosis in NSCLC patients.

Expression of FOXF1 was examined in an orthotopic mouse model of NSCLC using Lewis Lung Carcinoma (LLC) cells (Doki et al, 1999). LLC cells were injected into the left lung lobe of C57Bl/6 wild-type (WT) mice and mice were sacrificed 28 days after tumor inoculation. Expression of FOXF1 was visualized by

immunostaining for FOXF1 and SOX17, the latter is a known nuclear endothelial cell marker (Lange et al, 2014). Consistent with human data (Fig. 1A), the percentage of FOXF1-positive endothelial cells in mouse LLC tumors was decreased compared to normal lung tissue (Figs. 1F and EV1B). Next, endothelial cells were FACS-sorted from microdissected LLC tumors and compared with endothelial cells from control lungs. *Foxf1* mRNA was decreased in TEC compared to normal EC (Fig. 1G). Thus, both FOXF1 protein and mRNA expression were decreased in TEC of mouse lung tumors, recapitulating the human data.

To confirm our results using FACS-sorted lung endothelial cells and immunostaining, we next used a publicly available scRNA-seq dataset to analyze endothelial cells from LLC tumors and normal murine lung (Goveia et al, 2020). Endothelial cells from normal lung (NEC) and LLC tumors (TEC) were visualized using uniform manifold approximation and projection (UMAP) after samples integration with Harmony (Korsunsky et al, 2019) (Figs. 1H and EV1C). Both *FOXF1* mRNA levels and the total number of FOXF1-expressing EC were decreased in lung tumors compared to normal lung (Fig. 1H). Based on endothelial cell sub-clustering, decreased Foxf1 mRNA was detected in tumor-associated capillary, arterial and venous ECs (Fig. EV1D). Foxf1 was not expressed in the lymphatics (Fig. EV1D). Altogether, FOXF1 expression is decreased in TEC of mouse and human NSCLC tumors.

## Deletion of *Foxf1* in endothelial cells promotes lung tumor growth and metastasis

To determine the role of FOXF1 in endothelial cells during lung tumor progression, we used mice in which the *Foxf1* gene was deleted in endothelial cells (Cai et al, 2016). Since homozygous deletion of *Foxf1* in mouse endothelial cells (*Pdgfb-CreER*$^{tg/+}$;*Foxf1*$^{fl/fl}$) is lethal (Cai et al, 2016), we used heterozygous mice for cancer studies (*Pdgfb-CreER*$^{tg/+}$;*Foxf1*$^{fl/+}$; abbreviated as end*Foxf1*$^{+/-}$). Upon tamoxifen (Tam) administration, Cre-mediated recombination of the *Foxf1-flox* allele causes deletion of exon 1 encoding the DNA-binding domain of FOXF1 protein (Fig. EV2A,B). Single transgenic *Foxf1*$^{fl/+}$ or *Pdgfb-Cre*$^{tg/+}$ littermates were used as controls. Next, lung ECs were FACS-sorted from Tam-treated control and end*Foxf1*$^{+/-}$ mice, and *Foxf1* mRNA was assessed by qRT-PCR. The ~50% decrease in *Foxf1* mRNA was observed in ECs from end*Foxf1*$^{+/-}$ lungs compared to controls (Fig. EV2C,D). Lungs of Tam-treated control and *endFoxf1*$^{+/-}$ mice without tumors were histologically normal (Fig. EV2E), a finding consistent with published studies (Cai et al, 2016). Orthotopic LLC lung tumors in Tam-treated end*Foxf1*$^{+/-}$ mice developed faster and were significantly larger than in control mice (Fig. 2A–D). BrdU incorporation and immunostaining for phospho-histone H3 (pH3) showed increased numbers of proliferating tumor cells in end*Foxf1*$^{+/-}$ mice compared to controls (Appendix Fig. S1A,B). Using FACS-sorted lung endothelial cells, we found that TECs from end*Foxf1*$^{+/-}$ tumors had a more profound decrease of *Foxf1* mRNA compared to TECs from control tumors (Fig. 2E). Moreover, deletion of *Foxf1* in endothelial cells increased the number of mice with mediastinal metastases (Fig. 2F; Appendix Fig. S1C). To demonstrate that the role of FOXF1 in endothelial cells is not limited to LLC tumor model, we utilized a mouse model of urethane-induced lung carcinogenesis (Miller et al, 2003) (Fig. 2G). Similar to the orthotopic LLC model, Tam-treated end*Foxf1*$^{+/-}$ mice had increased numbers and sizes of tumors (Fig. 2H–J; Appendix Fig. S1D). Increased

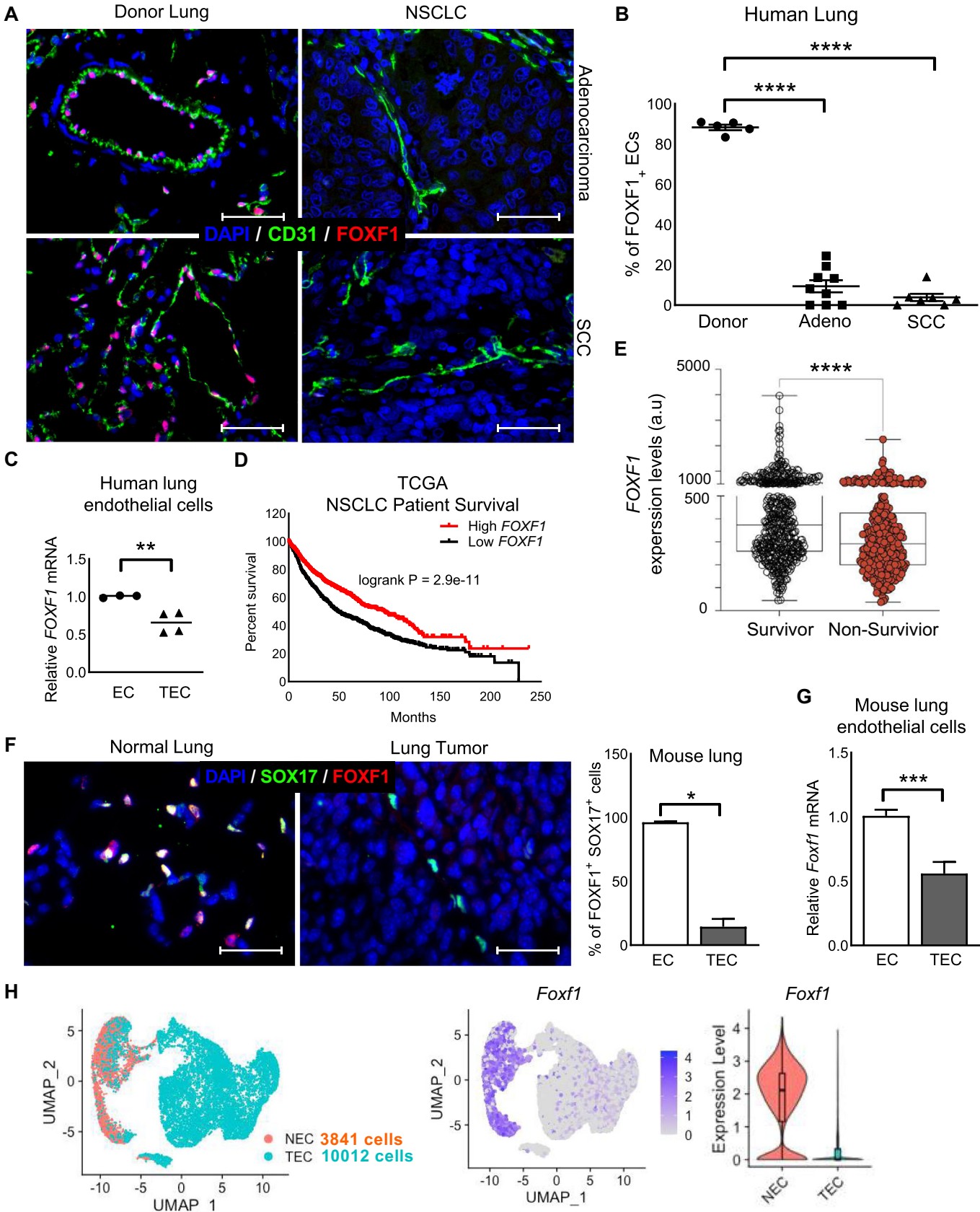

**Figure 1. FOXF1 is decreased in human and mouse NSCLC-associated endothelial cells.**

(A) Co-localization studies show decreased FOXF1 staining (red) in CD31+ endothelial cells (green) within human lung adenocarcinomas and lung squamous cell carcinomas (SCC) compared to endothelial cells of donor lung. (*N* = 5–9 per group). Scale bar = 50 μm. (B) Percentage of FOXF1+/CD31+ double-positive cells are decreased within lung adenocarcinomas and SCC compared to donor lungs. Five random fields were counted and presented as mean ± SEM. (*N* = 5–9 per group). ****P < 0.0001. (C) *FOXF1* mRNA is decreased in tumor-associated endothelial cells (TEC) FACS-sorted from NSCLC specimens compared to normal endothelial cells (EC) FACS-sorted from donor lungs. Endothelial cells were identified as CD31+/CD45− cells. Data presented as mean ± SEM. (EC, *N* = 3; TEC, *N* = 4). **P = 0.0087. (D, E) TCGA data mining show that lower *FOXF1* mRNA levels in tumors predict poor overall survival in NSCLC patients. Median *Foxf1* and *Fzd1* mRNA levels were used to split the two populations (*N* = 1926). ****P < 0.0001. Boxplots show median, Q1 and Q3 quartiles and whiskers up to 1.5× interquartile range. (F) Co-localization studies demonstrate decreased FOXF1 protein (red) in SOX17+ endothelial cells (green) within mouse LLC tumors compared to normal lungs. The percent of FOXF1+/SOX17+ double-positive cells were counted in five random fields and presented as mean ± SEM. (*N* = 4–5 mice per group). *P = 0.0159. (G) *Foxf1* mRNA is decreased in CD31+/CD45− cells isolated from mouse LLC tumors (TEC) compared to normal lungs (EC) as shown by qRT-PCR. β-actin mRNA was used for normalization. (*N* = 7–8 per group). ***P = 0.0005. (H) Unsupervised clustering of lung endothelial cells from scRNA-seq dataset (Goveia et al, 2020) is shown using uniform manifold approximation and projection (UMAP). Left panel: colored by group; red—Donor EC, (NEC; *n* = 3841 cells); blue—EC from LLC tumors, (TEC, *n* = 10012 cells). Middle panel: expression of *FOXF1* in NEC and TEC clusters after Z-score normalization. Right panel: violin plots show decreased expression of *FOXF1* mRNA in TECs compared NECs. *FOXF1* expression is log normalized. Boxplots show median, Q1 and Q3 quartiles and whiskers up to 1.5× interquartile range. Scale bars = 50 μm. Data information: Data represent different numbers (*N*) of biological replicates. The data with error bars are shown as mean ± SEM. Statistical analysis was performed using using the two-tailed unpaired-sample Student *t* test (C, E, F, G), logrank test (D), or one-way ANOVA followed by Tukey's post hoc test (B). Source data are available online for this figure.

tumorigenesis in end*Foxf1*+/− mice was associated with increased proliferation of tumor cells (Appendix Fig. S1E). Altogether, FOXF1 deficiency in endothelial cells stimulated lung tumorigenesis in orthotopic and chemically induced mouse models of lung cancer.

## Deletion of *Foxf1* in endothelial cells causes functional and structural abnormalities in tumor vasculature

We next examined tumor-associated angiogenesis in FOXF1-deficient mice. The number of tumor-associated vessels and *Pecam1* mRNA in microdissected LLC tumors were increased in end*Foxf1*+/− lungs compared to controls (Fig. 3A,B). Flow cytometry of microdissected tumors demonstrated the increased number of tumor-associated CD31+CD45− endothelial cells in end*Foxf1*+/− tumors compared to control tumors (Fig. 3C). The end*Foxf1*+/− tumor vessels had decreased pericyte coverage as demonstrated by immunostaining for endothelial marker CD31 and pericyte marker NG2 (Fig. 3D). Tumor-associated vasculature in end*Foxf1*+/− lungs had decreased collagen IV staining in basement membranes, and the ratio between CD31 and collagen IV was reduced (Fig. 3E), consistent with the loss of basement membrane. To evaluate the perfusion of tumor-associated blood vessels, Texas red-conjugated lectin was injected into the tail vein of tumor-bearing mice. Perfusion of tumor vessels in end*Foxf1*+/− lungs was decreased compared to tumors vessels in control lungs (Fig. 3F). To determine whether decreased perfusion impacted oxygen availability within the tumor, we injected tumor-bearing mice with pimonidazole, a chemical sensor for hypoxia (Cantelmo et al, 2016). LLC tumors in end*Foxf1*+/− mice had more hypoxic areas (Fig. 3G), a finding consistent with decreased vascular perfusion. Proliferation of endothelial cells in LLC lung tumors was low as determined by co-staining of lung sections with Ki-67 and CD31 (Appendix Fig. S2). Altogether, deletion of *Foxf1* in endothelial cells caused structural abnormalities in tumor-associated blood vessels, associated with decreased vascular perfusion and increased hypoxia in tumor tissue.

## Deletion of *Foxf1* decreases Wnt/β-catenin signaling in tumor-associated endothelial cells

To identify molecular mechanisms by which FOXF1 regulates TECs, we performed RNA-seq analysis using TECs isolated from microdissected LLC tumors of control and end*Foxf1*+/− mice. Gene

Set Enrichment Analysis of RNA-seq data showed that several biological processes and signaling pathways were altered in FOXF1-deficient TECs, including canonical Wnt/β-catenin signaling pathway, which was decreased in end*Foxf1*+/− TECs compared to control TECs (Fig. 4A, left panels). Furthermore, mRNAs of downstream targets of canonical Wnt/β-catenin pathway were decreased in end*Foxf1*+/− TECs. These include *Axin1, Axin2, Dvl2, Nkd1, Lef1, Sfrp2, Wnt4, Sfrp1, Plcb2,* and *Nkd2* mRNAs (Fig. 4A, right panel). Since Wnt/β-catenin signaling regulates vascular normalization in mouse glioma models (Reis et al, 2012), we examined nuclear localization of β-CATENIN in TECs. In control mice, nuclear β-CATENIN was detected in both TECs and tumor cells as shown by immunostaining for β-CATENIN and CD31 (Fig. 4B). In contrast, nuclear β-CATENIN was decreased in TECs of end*Foxf1*+/− mice (Fig. 4B,C). Consistent with decreased Wnt/β-catenin signaling pathway, *Axin2, Axin1* and *Lef1* mRNAs, known downstream targets of the Wnt/β-catenin pathway (Jho et al, 2002), were reduced in endothelial cells isolated from end*Foxf1*+/− tumors (Fig. 4D). In human pulmonary arterial endothelial cells (HPAEC), the efficient shRNA-mediated inhibition of *FOXF1* decreased *LEF1* mRNA (Fig. 4F) but increased growth of spheroids formed by human H-441 lung adenocarcinoma cells and FOXF1-deficient HPAECs (Fig. 4E). Nuclear β-CATENIN and downstream targets of WNT/β-catenin pathway was decreased in TECs within human lung adenocarcinomas and squamous cell carcinomas (SCC) (Fig. 4G–I), coinciding with reduced FOXF1 expression in TECs (Fig. 1A–C). Thus, FOXF1 deficiency is associated with decreased Wnt/β-catenin signaling in pulmonary endothelial cells in mouse and human NSCLC tumors.

## Endothelial-specific expression of FOXF1 prevents lung tumor progression in vivo

As depletion of *Foxf1* in endothelial cells accelerated cancer progression in orthotopic and urethane-induced lung cancer models (Fig. 2), we next tested whether increasing the *Foxf1* expression in ECs would prevent lung tumor progression and metastasis. We generated triple transgenic mice (*Pdgfb-CreER*tg/+; *LSL-rtTA*tg/+; *TetO-Foxf1*tg/+) with conditional expression of *Foxf1* in endothelial cells, abbreviated here as end*Foxf1*OE (Fig. EV3A). After treatment with tamoxifen, rtTA is expressed in endothelial

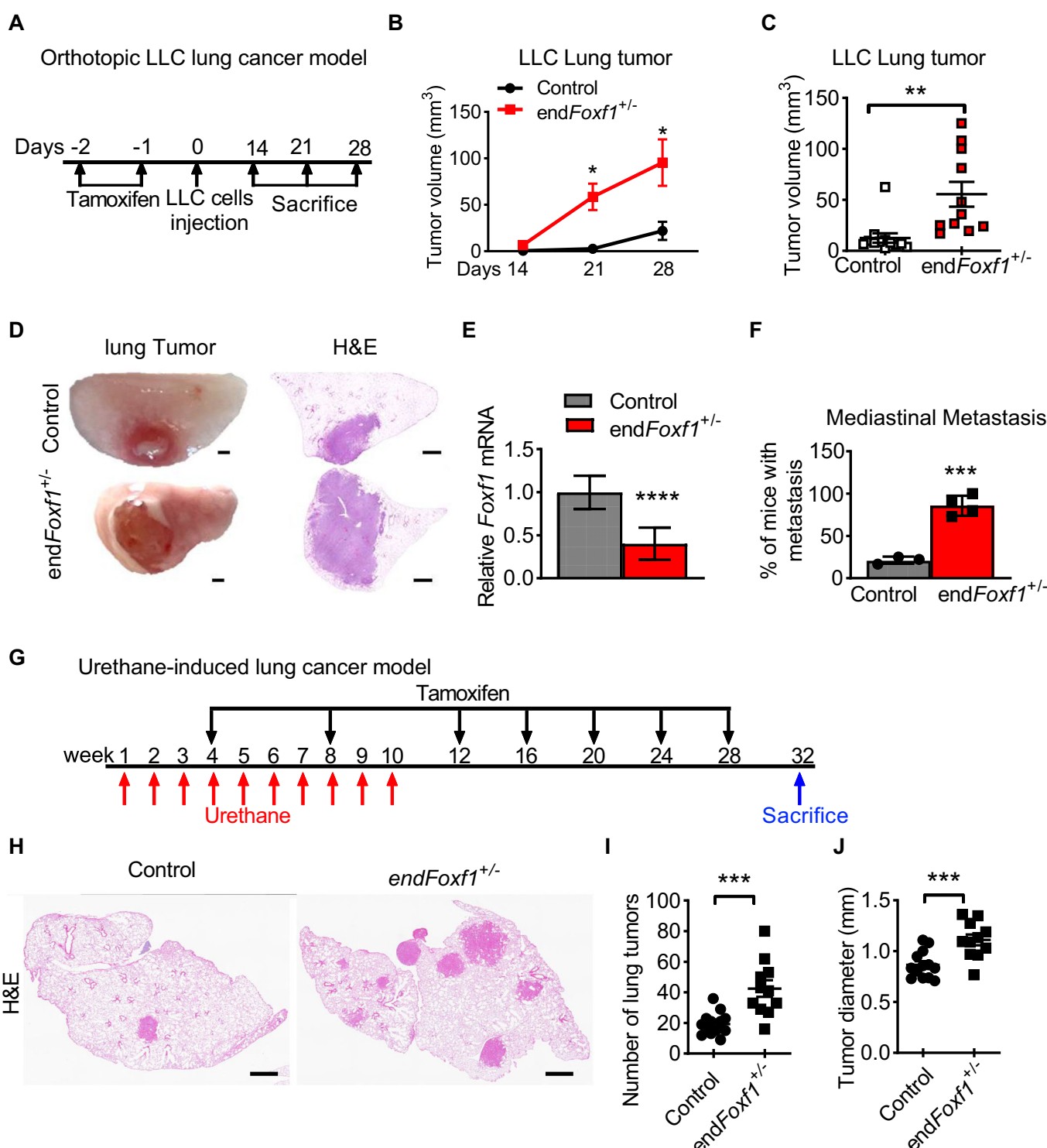

**A** Orthotopic LLC lung cancer model

**B** LLC Lung tumor

**C** LLC Lung tumor

**D** lung Tumor  H&E

**E** Relative *Foxf1* mRNA

**F** Mediastinal Metastasis

**G** Urethane-induced lung cancer model

**H** H&E  Control  *endFoxf1*⁺/⁻

**I** Number of lung tumors

**J** Tumor diameter (mm)

cells. After treatment with doxycycline, rtTA-mediated activation of TetO promoter causes an increase in *Foxf1* expression (Fig. EV3A,B). Altogether, treatment with tamoxifen+doxycycline increases FOXF1 expression, specifically in endothelium. Endothelial cells isolated from lungs of tamoxifen+doxycycline-treated endFoxf1^OE mice exhibited increased *Foxf1* mRNA when compared to single transgenic controls (Fig. EV3F). The endFoxf1^OE mice had

no visible phenotype with normal lung architecture (Fig. EV3C), and the expression of endothelial and pericyte markers was normal (Fig. EV3D,E). LLC tumor cells were inoculated into lungs of control and endFoxf1^OE mice. At 28 days after inoculation, tumors in the endFoxf1^OE mice were smaller (Fig. 5A–C) and the incidence of mediastinal metastasis decreased (Fig. 5D). FOXF1 immunostaining and *Foxf1* mRNA were increased in TECs from endFoxf1^OE

◄ **Figure 2.   Deletion of *Foxf1* in endothelial cells promotes lung tumor growth and metastasis.**

(A) Schematic representation of LLC orthotopic model of lung cancer. Tamoxifen-treated mice were inoculated with mCherry-labeled LLC cells into the left lung lobe. Mice were sacrificed on days 14, 21, and 28 post LLC injection (day 21: *P = 0.0291; D28: *P = 0.0232). (B) Growth of lung tumors in end*Foxf1*$^{+/-}$ mice is increased compared to controls. (N = 8–11 mice per group). (C) Deletion of *Foxf1* in endothelial cells increased lung tumor volume measured on day 28. (N = 11–12 mice per group). **P = 0.0026. (D) H&E staining shows larger LLC tumors in end*Foxf1*$^{+/-}$ mice compared to controls. Scale bar = 500 μm. (E) qRT-PCR shows that *Foxf1* mRNA is decreased in endothelial cells FACS-sorted from microdissected tumors of end*Foxf1*$^{+/-}$ mice as compared to control mice. β-*actin* mRNA was used for normalization. (N = 7 mice per group). ****P < 0.0001. (F) Tumor-bearing end*Foxf1*$^{+/-}$ mice have increased incidence of mediastinal lymph node metastasis. (N = 3–4 mice per group). ***P = 0.0003. (G) Schematic representation of urethane-induced lung cancer model. Tamoxifen-treated end*Foxf1*$^{+/-}$ and control mice were administered urethane once a week for 10 weeks. Mice were sacrificed 32 weeks post first urethane injection. (H) H&E staining shows increased tumor number and size in urethane-treated end*Foxf1*$^{+/-}$ mice. Scale bar = 500 μm. (I, J) Deletion of *Foxf1* in endothelial cells increased the number and diameter of lung tumors in urethane-treated end*Foxf1*$^{+/-}$ mice. (N = 11–13 mice per group). (I: ***P = 0.0003; J: ***P = 0.0009). Data information: Data represent different numbers (N) of biological replicates. The data with error bars are shown as mean ± SEM. Statistical analysis was performed using the two-tailed unpaired-sample Student t test (C, E, F, I, J), or two-way ANOVA followed by Fisher's LSD post hoc test (B). Source data are available online for this figure.

tumors (Fig. 5E,F). Increased pericyte coverage and improved basement membrane were evident in end*Foxf1*$^{OE}$ tumor vessels as shown by co-localization of CD31 with either NG2 or Collagen IV (Fig. 5F–H). The percentage of endothelial cells with nuclear β-CATENIN was increased in lung tumors of end*Foxf1*$^{+/-}$ mice (Fig. 5I), coinciding with increased *Axin2* mRNAs in flow-sorted endothelial cells (Fig. 5F). Altogether, increased expression of FOXF1 in endothelial cells stimulates canonical WNT/β-catenin signaling, stabilizing tumor-associated blood vessels, and decreasing lung tumor growth and metastasis.

## FRIZZLED-4 is decreased after deletion of *Foxf1* in lung endothelial cells

To investigate molecular mechanisms by which FOXF1 regulates Wnt/β-catenin signaling in TECs, we examined the expression of Frizzled receptors. Based on the scRNA-seq analysis, the highest level of mRNA expression in all subtypes of lung endothelial cells was observed for *Fzd4* mRNA (Fig. EV4A). Approximately 20% of non-capillary ECs expressed *Fzd6*, and the expression levels of *Fzd1, Fzd2, Fzd3, Fzd5, Fzd7, Fzd8, Fzd9*, and *Fzd10* were undetectable (Fig. EV4A). Using RNA-seq analysis and qRT-PCR of FACS-sorted TECs, we found that deletion of *Foxf1* decreased *Fzd4* mRNA in TECs (Fig. 6A,B). Furthermore, reduced FZD4 staining was detected in TECs from end*Foxf1*$^{+/-}$ mice compared to TECs from control mice and normal mouse lung (Fig. 6C). In human NSCLC, FZD4 staining in TECs was decreased compared to endothelial cells from donor's lungs (Fig. 6D). TCGA data mining showed that NSCLC patients with higher *FZD4* expression had increased overall survival compared to patients with lower *FZD4* expression (Fig. 6E), coinciding with similar correlation between FOXF1 and patient survival (Fig. 1D). Finally, in human pulmonary arterial endothelial cells (HPAEC) in vitro, the efficient shRNA-mediated inhibition of *FOXF1* decreased *FZD4* (Fig. 4F). Altogether, inhibition of FOXF1 in endothelial cells in vivo decreases Wnt/β-catenin signaling and reduces FZD4.

## FOXF1 directly binds to and induces the transcriptional activity of *Fzd4* promoter

We next determined whether FOXF1 directly regulates the transcription of the *Fzd4* gene. A previously published ChIP-seq dataset (Dharmadhikari et al, 2016) was used to demonstrate that FOXF1 directly bound to the DNA regulatory element located at 3'

to exon 2 of the *Fzd4* gene in mouse endothelial cells (Figs. 6F and EV4B). Within the FOXF1-binding region, several FOXF1 consensus DNA sequences were identified (Fig. EV4B). To test whether FOXF1 directly activates *Fzd4* transcription, we cloned the *Fzd4* regulatory element into the pGL2-basic luciferase (LUC) vector to generate *Fzd4-Luc* reporter containing 4 endogenous FOXF1-binding sites, BDS1, 2, 3, and 4 (Fig. 6G). Next, we used site-directed mutagenesis to sequentially disrupt each FOXF1-binding site and generate four mutant Fzd4-Luc constructs (Fig. 6G). The Fzd4-Luc reporter or its mutants were co-transfected with either CMV-empty or CMV-*Foxf1* expression plasmids and the transcriptional activation of *Fzd4* DNA region was tested by measuring LUC activity. Co-transfection with the CMV-*FOXF1* plasmid increased transcriptional activity of the *Fzd4* regulatory element containing all four FOXF1-binding sites (Fig. 6G). Disruption of BDS3 completely abrogated the ability of FOXF1 to stimulate Fzd4 transcriptional activity, whereas disruption of BDS1 or BDS4 had partial effects (Fig. 6G). Disruption of BDS2 had no effect. Thus, FOXF1 binds to and induces transcriptional activity of the *Fzd4* regulatory element through BDS1, 3, and 4, indicating that *Fzd4* is a direct target of FOXF1.

## Nanoparticle delivery of FZD4 into endothelial cells decreases lung tumor growth in FOXF1-deficient mice

To determine whether FOXF1 acts via FZD4, we tested whether nanoparticle delivery of *Fzd4* cDNA into endothelial cells will decrease lung tumor growth in FOXF1-deficient mice. Mouse *Fzd4* cDNA was cloned into CMV vector to generate the CMV-*Fzd4* construct. To deliver CMV-*Fzd4* plasmid to endothelial cells in vivo, we utilized the poly β-amino esters (PBAE) polymer (Kaczmarek et al, 2018; Kaczmarek et al, 2021), which forms stable nanoparticles in complex with plasmid DNA and is suitable to deliver DNA plasmids to endothelial cells in vivo (Bian et al, 2023). Fluorescently labeled PBAE nanoparticles with either CMV-*Fzd4* (nano-*Fzd4*) or CMV-Empty (nano-Empty) plasmids were delivered intravenously to tumor-bearing mice one week after implantation of LLC tumor cells (Fig. 7A). To test the safety of nanoparticles, we assessed liver and kidney functions. Mice treated with nanoparticles did not show any differences in liver and kidney metabolic panels compared to controls, including levels of total protein, Albumin, Globulins, ALP, Total Bilirubin, GGT, ALT, BUN and Creatinine (Fig. EV5A; Appendix Table S4). Also, no changes in hematologic parameters were found in the nano-Fzd4

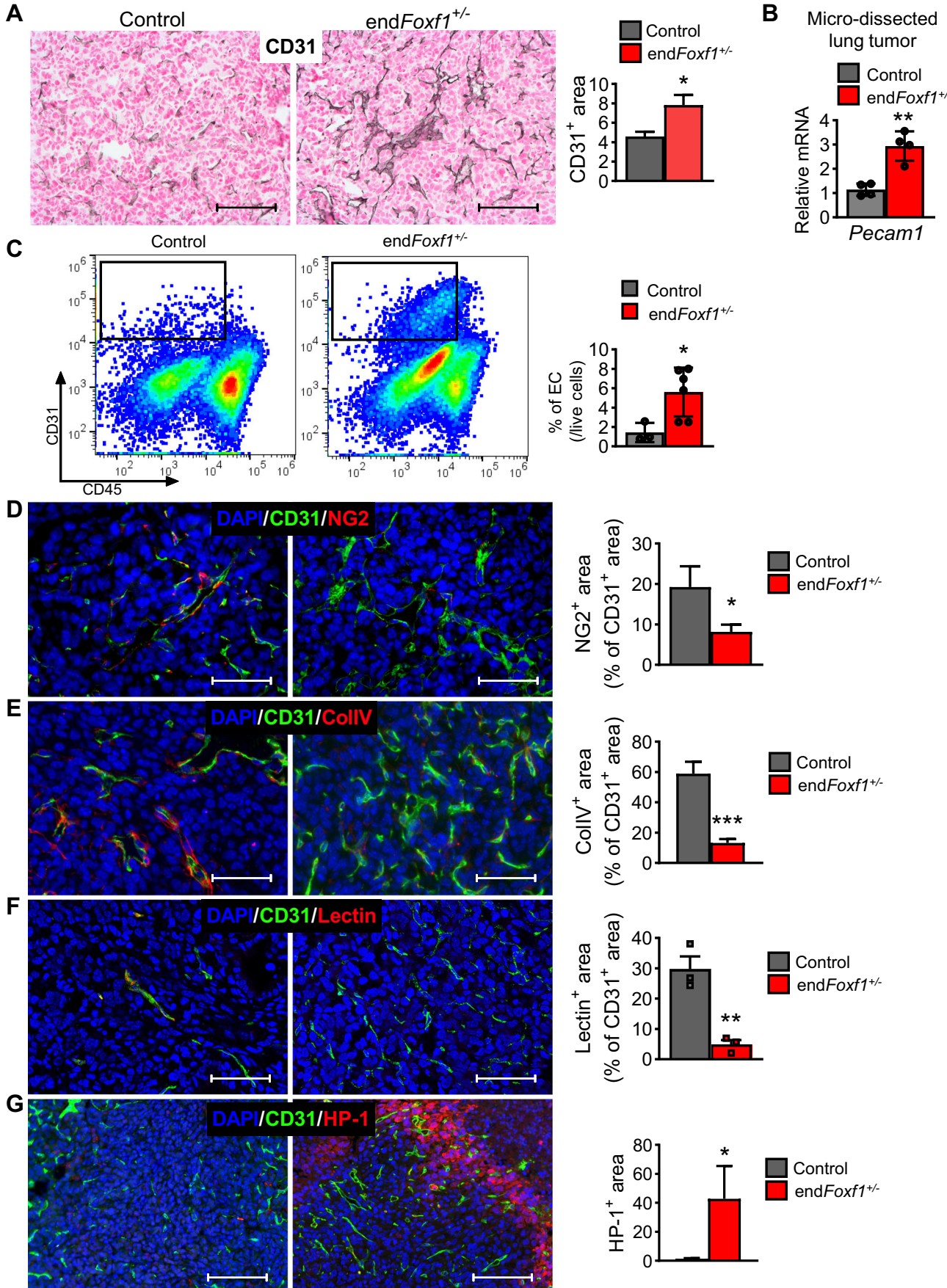

**Figure 3. Deletion of *Foxf1* in endothelial cells causes functional and structural abnormalities in tumor vasculature.**

(A) The density of tumor vessels is increased in LLC tumors of end*Foxf1*[+/−] mice as shown by immunostaining with CD31[+] Abs. ($N = 9$ mice per group). *$P = 0.0123$. (B) qRT-PCR shows that *Pecam1* mRNA is increased in microdissected lung tumors from end*Foxf1*[+/−] mice compared to control mice. β-*actin* mRNA was used for normalization. ($N = 4$ per group). **$P = 0.0016$. (C) Increased percent of endothelial cells in microdissected lung tumors from end*Foxf1*[+/−] mice is shown by flow cytometry. ($N = 3–6$ mice per group). *$P = 0.0317$. (D) Tumor-associated vessels in end*Foxf1*[+/−] lungs have decreased pericyte coverage as demonstrated by immunostaining for endothelial marker CD31 (green) and pericyte marker NG2 (red). The percentage of positive area was counted in five random fields and presented as mean ± SEM. ($N = 8–9$ mice per group). *$P = 0.0491$. (E) The end*Foxf1*[+/−] tumor vessels have decreased basement membrane, shown by decreased Collagen IV immunostaining (red), and reduced ratio between CD31 (green) and Collagen IV levels (red). The percentage of positive area was counted in five random fields and presented as mean ± SEM. ($N = 5–6$ mice per group). **$P = 0.0043$. (F) Perfusion of tumor vessels in endFoxf1[+/−] lungs is decreased compared to tumor vessels in control lungs. Texas red-conjugated lectin was injected into the tail vein of tumor-bearing mice. Perfusion is shown by co-staining for CD31[+] (green) blood vessels and tdTomato-lectin (red) ($N = 3$ mice per group). **$P = 0.0051$. (G) LLC tumors in endFoxf1[+−] lungs have more hypoxic areas as shown using hypoxyprobe (red). Tumor vessels are visualized using immunofluorescence for CD31[+] (green). The percentage of positive area was counted in five random fields and presented as mean ± SEM. ($N = 5–6$ mice per group). **$P = 0.0043$. Scale bar = 50 μm. Data information: Data represent different numbers ($N$) of biological replicates. The data with error bars are shown as mean ± SEM. Statistical analysis was performed using the two-tailed unpaired-sample Student *t* test (A–G). Source data are available online for this figure.

treated group compared to the control group (Fig. EV5B; Appendix Table S5). Furthermore, no significant body weight changes were found between nano-Fzd4-treated and control groups (Fig. EV5C). Finally, the nano-Fzd4 treatment does not change the histological appearance of endothelial cells in the liver, as demonstrated by immunostaining for Pecam1 (CD31) (Fig. EV5D). Altogether, the treatment of mice with nano-Fzd4 was not toxic. Using FACS analysis of enzymatically digested lung tissue, nanoparticles were detected in ~70% of lung endothelial cells (CD31[+]/CD45[−]) (Fig. 7B). Nanoparticle-mediated targeting of epithelial (CD326[+]/CD45[−]/CD31[−]), hematopoietic/immune (CD45[+]/CD31[−]) and mesenchymal (CD31[−]/CD45[−]/CD326[−]) cells was ineffective (Fig. 7B). Treatment with nano-*Fzd4* significantly decreased lung tumor sizes in end*Foxf1*[+/−] mice compared to nano-Empty controls (Fig. 7C,D). Decreased tumor sizes in EEV-*Fzd4*-treated mice were associated with increased number of TECs expressing *Fzd4 mRNA* (Fig. 7E) as demonstrated by RNAscope with probes specific to *Fzd4*, *Foxf1*, and *Aplnr*, the latter of which is a specific marker of general capillary cells in the lung (Gillich et al, 2020). The number of TECs expressing nuclear β-catenin protein were also increased compared to nano-Empty-treated tumor-bearing end*Foxf1*[+/−] mice (Fig. 7F), consistent with increased Wnt/β-catenin signaling in TECs. In addition, nanoparticle delivery of *Fzd4* to endothelial cells increased protein levels of CollV, which is the major protein in lung endothelial basal membrane (Fig. 7G) and improved the expression of Claudin-5 and VE-cadherin that were impaired in TECs (Appendix Fig. S3). Altogether, nanoparticle delivery of *Fzd4* cDNA into endothelial cells of tumor-bearing end*Foxf1*[+/−] mice restored Fzd4 expression, increased nuclear β-catenin, improved endothelial basal membrane, and inhibited lung tumor growth. Thus, FOXF1 stimulates Wnt/β-catenin signaling in endothelial cells through FZD4. Nanoparticle delivery of Fzd4 into TECs can be a therapeutic approach to inhibit lung tumor growth.

## Discussion

Lung cancer is the leading cause of cancer-related mortality worldwide. Despite the promise of antiangiogenic therapies in animal models of cancer, their success has been limited in clinical trials to treat advanced NSCLC. Bevacizumab is a humanized anti-VEGF monoclonal antibody that binds to soluble VEGF and prevents the activation of VEGFR2 receptor (Presta et al, 1997).

The combination therapy of bevacizumab with carboplatin and paclitaxel showed only modest improvement in overall survival and was associated with significant toxicities (Soria et al, 2013). Therefore, additional approaches targeting tumor-associated vasculature are needed for NSCLC therapies.

This study supports the concept that increasing expression of FOXF1 or its downstream target FZD4 in tumor endothelial cells, either pharmacologically or via gene therapy, can be considered for NSCLC treatment. FOXF1 is a transcription factor expressed in pulmonary endothelial cells, which has been extensively studied in embryonic and postnatal lung development (Kalinichenko et al, 2001b). FOXF1 is required for formation of blood vessels by activating VEGF, PDGF-β and TIE-2 signaling pathways (Ren et al, 2014). Deletion and inactivating point mutations in *FOXF1* gene locus are associated with Alveolar Capillary Dysplasia with Misalignment of Pulmonary Veins (ACDMPV), a congenital pulmonary disorder characterized by the loss of pulmonary capillaries and respiratory insufficiency in neonates and infants (Dharmadhikari et al, 2015). FOXF1 stimulates lung repair and regeneration in adult mice by regulating genes critical for extracellular matrix remodeling, inflammation, and endothelial barrier function (Bolte et al, 2017; Cai et al, 2016; Kalin et al, 2008). Consistent with these studies, we found that FOXF1 stabilizes blood vessels inside the NSCLC tumors, which can be utilized to enhance delivery of chemotherapeutic agents or immune checkpoint inhibitors in NSCLC therapy. In addition to its vessel-stabilizing property, FOXF1 was implicated in embryonic angiogenesis (Pradhan et al, 2019; Ren et al, 2014; Sturtzel et al, 2018; Sun et al, 2021), anti-fibrotic effects in lung fibroblasts (Black et al, 2018), supporting lung regeneration after pneumonectomy (Bolte et al, 2017; Cai et al, 2016), and engraftment of endothelial progenitor cells after cell therapy (Kolesnichenko et al, 2023; Wang et al, 2021). Furthermore, FOXF1 was found to be required for oncogenic potential of tumor cells in rhabdomyosarcomas (Milewski et al, 2017; Milewski et al, 2021) and gastrointestinal stromal tumors (Ran et al, 2018). These published studies demonstrate that FOXF1 plays divers roles in different diseases by regulating multiple downstream signaling pathways in cell-specific manner reviewed in (Bolte et al, 2020a) and (Bolte et al, 2018). While therapeutic targeting of FOXF1 can be achieved using either gene therapy (Bolte et al, 2020b) or TanFe small molecule compound (Pradhan et al, 2023), further understanding of cell- and context-dependent functions of FOXF1 is needed for future considerations of FOXF1-directed therapies.

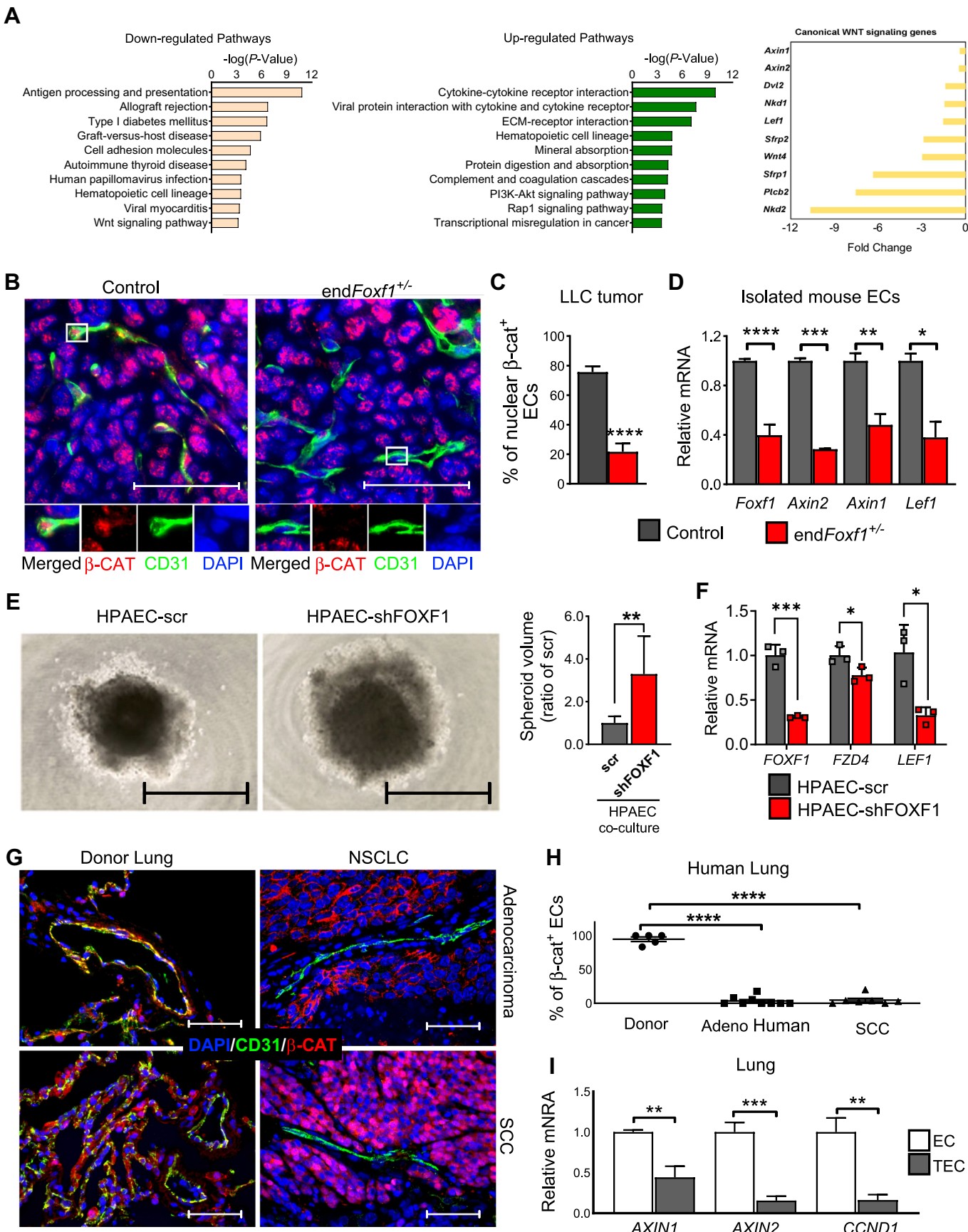

**Figure 4. Deletion of *Foxf1* in tumor endothelial cells decreases canonical WNT signaling.**

(A) Gene Set Enrichment Analysis of RNA-seq data shows downregulated pathways, including canonical Wnt signaling in end*Foxf1*+/− compared to control TECs (left panels). Endothelial cells isolated from microdissected end*Foxf1*+/− and control tumors were used for RNA-seq. Reduction in gene transcripts related to canonical WNT signaling pathway is presented as fold changes between end*Foxf1*+/− and control TECs (right panel). (B) Co-localization studies show decreased nuclear β-CATENIN (red) staining in CD31$^+$ (green) endothelial cells within LLC tumors of end*Foxf1*$^{+/-}$ mice. Scale bar = 50 μm. (C) Percent of β-CATENIN$^+$/CD31$^+$ double-positive cells were counted in five random fields and presented as mean ± SEM. ($N$ = 5–7 mice per group). ****$P$ < 0.0001. (D) Deletion of *Foxf1* in endothelial cells decreases *Foxf1* mRNA as well as mRNAs of downstream targets of canonical WNT signaling pathway, including *Axin2*, *Axin1*, and *Lef1*. qRT-PCR was performed using FACS-sorted CD31$^+$/CD45$^-$ endothelial cells from LLC tumors of end*Foxf1*$^{+/-}$ and control mice. β-*actin* was used for normalization. ($N$ = 2–7 per group). (*Foxf1*: ****$P$ < 0.0001; *Axin2*: ***$P$ = 0.0009; *Axin1*: **$P$ = 0.0085; *Lef1*: *$P$ = 0.0116). (E) The growth of human H-441 lung tumor spheroids were decreased when co-cultured with FOXF1-deficient HPAEC compared to control HPAEC. Scale bar = 1000 μm. ($N$ = 9). **$P$ = 0.0015. (F) qRT-PCR shows the efficient shRNA-mediated inhibition of *FOXF1*, *FZD4*, and *LEF1* in human pulmonary arterial endothelial cells (HPAEC). β-*ACTIN* mRNA was used for normalization. ($N$ = 3). (*FOXF1*: ***$P$ = 0.0005; *FZD4*: *$P$ = 0.0433; *LEF1*: *$P$ = 0.0193). (G) Co-localization studies show decreased β-CATENIN (red) staining in CD31$^+$ (green) endothelial cells within human lung adenocarcinomas and SCC compared to donor lungs. Scale bar = 50 μm. (H) Percent of β-CATENIN$^+$/CD31$^+$ double-positive cells were counted in five random fields and presented as mean ± SEM. ($N$ = 5–9 per group). Scale bar = 50 μm. ****$P$ < 0.0001. (I) Downstream targets of canonical WNT signaling, including *AXIN2*, *AXIN1*, and *CCND1*, are decreased in TEC isolated from NSCLC patients compared to EC isolated from human donor lungs. β-*actin* mRNA was used for normalization. ($N$ = 3–5 per group). (*AXIN1*: **$P$ = 0.0052; *AXIN2*: ***$P$ = 0.0009; *CCND1*: **$P$ = 0.0017). Data information: Data represent different numbers ($N$) of biological replicates. The data with error bars are shown as mean ± SEM. Statistical analysis was performed using Benjamini-corrected $P$ value (A), the two-tailed unpaired-sample Student $t$ test (C, D, E, F, I), or one-way ANOVA followed by Tukey's post hoc test (H). Source data are available online for this figure.

Tumor blood vessels are characterized by poor perfusion, causing hypoxia and acidosis within the tumors (Semenza, 2013). Hypoxia increases tumor growth and prevents infiltration of anti-tumor cytotoxic T cells (Jayaprakash et al, 2018). Inhibition of tumor hypoxia decreases tumor growth and metastasis by increasing infiltration of CD8 T cells (Hatfield et al, 2015). Consistent with previous studies, we found that FOXF1-deficient endothelial cells form leaky tumor vasculature with abnormal basement membrane lacking ColIV, leading to hypoxia in the lung tumor. Our data suggests that FOXF1 expression in endothelial cells is required for vascular normalization and that increasing FOXF1 in TECs can potentially improve anti-tumor inflammatory responses and drug deliveries. Interestingly, we have observed elevated endothelial cell numbers in FoxF1-deficient tumors (Fig. 3), suggesting an increased tumor angiogenesis. This contrasts with diminished angiogenesis in FoxF1-deficient embryonic and neonatal lungs (Pradhan et al, 2019; Ren et al, 2014; Sturtzel et al, 2018; Sun et al, 2021). One explanation is that FOXF1 regulates angiogenesis differently in normal embryonic/neonatal lung tissue compared to lung tumor tissue. This is consistent with the different gene expression profiles in normal ECs (NEC) vs tumor-associated ECs (TEC) as reported in the present manuscript. Published studies demonstrate that FOXF1 can function as a pioneer transcription factor to remodel chromatin and make it accessible for other transcription factors to regulate gene expression (Ran et al, 2018). Therefore, it is possible that FOXF1-regulated chromatin remodeling can lead to either stimulation or inhibition of angiogenesis depending on other transcription factors that differentially expressed in normal and cancer lung tissues.

Canonical Wnt/β-catenin signaling plays a crucial role in endothelial cell proliferation, migration, and survival by activating Frizzled receptors. Activation or inhibition of β-catenin in endothelial cells causes embryonic lethality due to disruption of Wnt/β-catenin signaling and abnormal vessel adherens junctions (Haegel et al, 1995; Rudloff and Kemler, 2012). Activation of Wnt/β-catenin signaling in endothelial cells stabilizes postnatal retinal blood vessels (Birdsey et al, 2015), and improves stability of blood vessels in Erg-deficient mice (Birdsey et al, 2015). Interestingly, activation of Wnt/β-catenin signaling in endothelial cells inhibits glioma angiogenesis and normalizes tumor blood vessels through the recruitment of pericytes (Reis et al, 2012). Our studies support the concept that FOXF1 regulates vascular normalization within the lung tumor, at least in part, through transcriptional activation FZD4. In addition to its role in tumor vasculature, Wnt/β catenin signaling regulates stromal and tumor cells to promote carcinogenesis (reviewed in (Nusse and Clevers, 2017)). Furthermore, FOXF1 was implicated in the regulation of Wnt/β catenin signaling in fibroblasts and osteoblasts (Reza et al, 2023; Shen et al, 2020). Therefore, it is possible that targeting of endothelial-FOXF1-FZD4-Wnt//β catenin axis can be insufficient to effectively repress tumor growth. The use of multiple therapeutic agents simultaneously inhibiting tumor cells and cells of tumor microenvironment may be needed to supplement the endothelial-specific nanoparticle FZD4 therapy for NSCLC.

In summary, FOXF1 is decreased in tumor-associated endothelial cells in human and mouse NSCLC. Endothelial cell-specific deletion of *Foxf1* stimulates lung cancer growth and metastasis, whereas FOXF1 overexpression in endothelial cells inhibits lung tumorigenesis. FOXF1 induces Wnt/β-catenin signaling in endothelial cells through transcriptional activation of FZD4. FOXF1 regulates pericyte coverage and perfusion of tumor-associated blood vessels, supporting the use of FOXF1-activating therapies for vascular normalization in NSCLC patients.

# Methods

## Transgenic mice

*Generation of endothelial cell-specific Foxf1 heterozygous mice*: We have previously generated *Foxf1*$^{fl/fl}$ mice (Ren et al, 2014) and bred them into the C57Bl/6 mice (The Jackson Laboratory). *Foxf1*$^{fl/fl}$ mice were crossed with *Pdgfb-iCreER*$^{tg/-}$ transgenic mice (Claxton et al, 2008) received from Dr. Marcus Fruttiger (University College London) to generate *Pdgfb-iCreER/Foxf1*$^{fl/+}$ (abbreviated as end*Foxf1*$^{+/-}$) mice. To delete 1 allele of *Foxf1* from endothelial cells, tamoxifen (3 mg; Sigma) was given as oral gavage to 6–8 weeks old mice. *Generation of endothelial cell-specific FOXF1 over-expression mice*: We have recently generated *TetO7-HA-mFoxf1*$^{tg/+}$ mice (Bian et al, 2023). *TetO7-HA-mFoxf1*$^{tg/+}$ mice were crossed

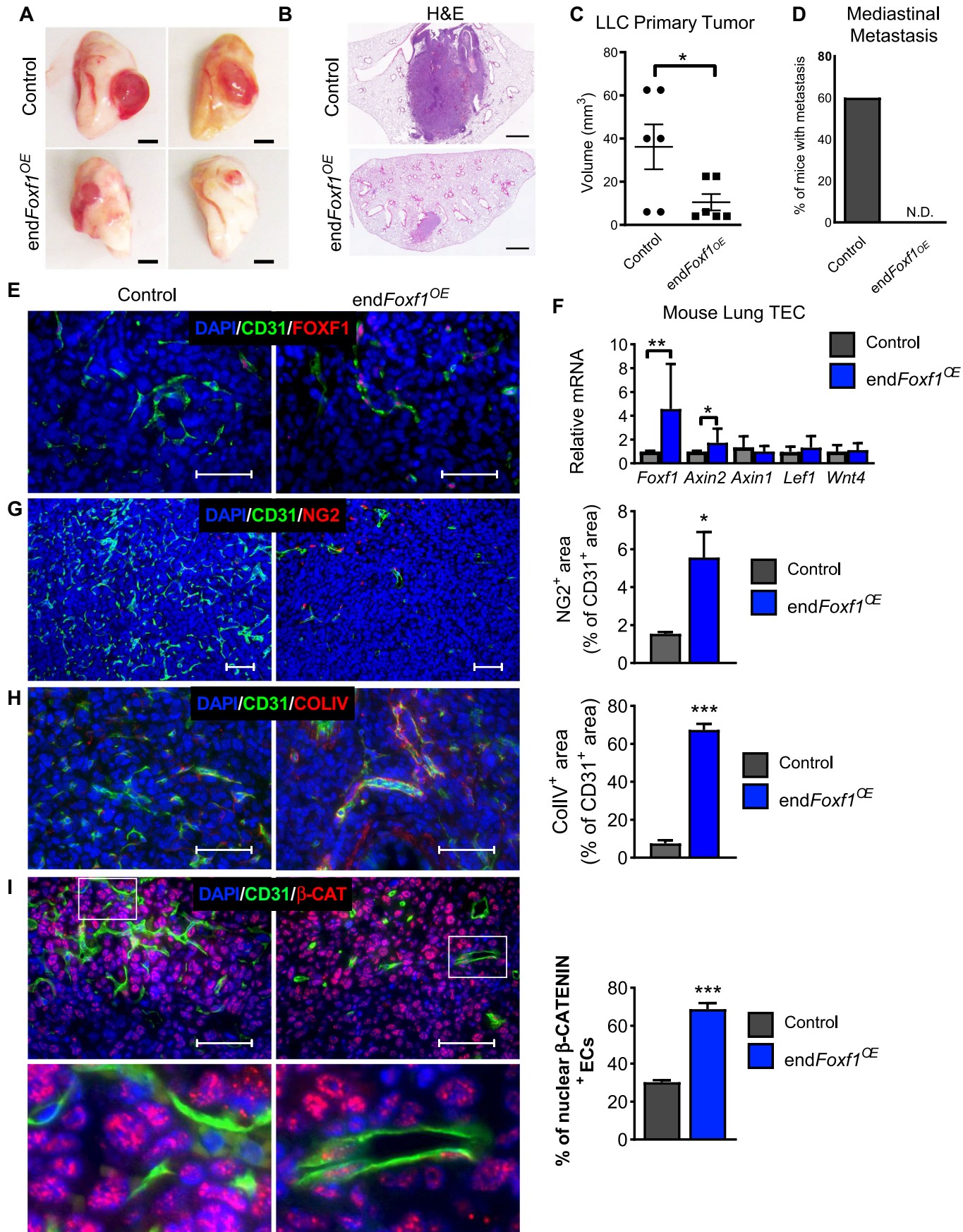

◄ **Figure 5. Endothelial-specific overexpression of FOXF1 inhibits lung cancer growth and metastases.**

(A) Overexpression of FOXF1 in endothelial cells (endFoxf1$^{OE}$) decreases primary lung tumor growth in LLC orthotopic mouse model. Scale bar = 0.2 cm. (B) H&E staining shows smaller LLC tumors in endFoxf1$^{OE}$ lungs compared to controls. Scale bar = 500 μm. (C, D) Overexpression of FOXF1 decreases lung tumor volume and frequency of mediastinal lymph node metastasis in endFoxf1$^{OE}$ mice. (N = 6 mice per group). (C: **P = 0.0428) (E) FOXF1 protein (red) is expressed in CD31$^+$ (green) endothelial cells within endFoxf1$^{OE}$ lung tumors but not within control lung tumors. (F) Foxf1 and Axin2 mRNAs are increased in CD31$^+$/CD45$^-$ TECs isolated from endFoxf1$^{OE}$ tumors compared to TECs from control tumors as shown by qRT-PCR. β-actin mRNA was used for normalization. (N = 5–12 per group). (Foxf1: **P = 0.0039; Axin2: *P = 0.0488). (G) Blood vessels in endFoxf1$^{OE}$ tumors have increased pericyte coverage as shown by co-localization of CD31 (green) with NG2 (red). The percentage of positive area was counted in five random fields and presented as mean ± SEM. (N = 3 mice per group). Scale bar = 50 μm. *P = 0.041. (H) Blood vessels in endFoxf1$^{OE}$ tumors have improved basement membrane as shown by increased co-localization of CD31 (green) with Collagen IV (red). The percentage of positive area was counted in five random fields and presented as mean ± SEM. (N = 3 mice per group). Scale bar = 50 μm. ****P < 0.0001. (I) The number of endothelial cells (green) with nuclear β-CATENIN (red) is increased in lung tumors of endFoxf1$^{+/-}$ mice. Percent of β-CATENIN$^+$/CD31$^+$ double-positive cells were counted in five random fields and presented as mean ± SEM. (N = 3 mice per group). Scale bar = 50 μm. ***P = 0.0003. Data information: Data represent different numbers (N) of biological replicates. The data with error bars are shown as mean ± SEM. Statistical analysis was performed using the two-tailed unpaired-sample Student t test (C, F, G, H, I). Source data are available online for this figure.

with *Pdgfb-iCreER*$^{tg/-}$ (Claxton et al, 2008) transgenic mice and *Rosa26*-LSL-rtTA$^{tg/+}$ (Belteki et al, 2005) (Jackson Laboratory) to generate *Pdgfb-iCreER/Rosa26*-LSL-rtTA/*TetO7-HA*-m*Foxf1* mice. To induce FOXF1 overexpression in endothelial cells, tamoxifen (3 mg; Sigma) was given as oral gavage on 2 consecutive days to 6–8 weeks old mice, followed by maintenance of mice on doxycycline chow for the duration of the experiment. Mice were sacrificed and lungs were collected 31 days post tamoxifen administration.

## Cell lines

Mouse Lewis Lung Carcinoma (LLC; ATCC # CRL-1642) were cultured in DMEM (Invitrogen), human lung adenocarcinoma cell line NCl-H-441 (ATCC, #HTB-174) were cultured in RPMI-1640. Human Pulmonary Artery Endothelial Cells (HPAEC, Lonza #CC2530) were cultured in ECM medium (ScienCell). For stable knockdown of *FOXF1*, HPAECs were transduced with TRC lentiviral human FOXF1 shRNA (clone ID: TRCN0000013953, Horizon). GIPZ non-silencing lentiviral shRNA (Cat #: RHS4346, Horizon) was used as described (Bian et al, 2023). Cells were tested mycoplasma-negative prior to experiments.

## Three-dimensional (3D) co-culture spheroid model

In total, 50 μl of H-441 cells (6 × 10$^6$ cells/ml) mixed with 50 μl of HPAEC cells (6 × 10$^6$ cells/ml) and seeded to 96-well ultra-low attachment 96-well plates (Corning, cat# CLS7007). The plates were then placed in the incubator at 37 °C with 5% CO$_2$, and the spheroids grew on the bottom of the wells. After 7-day incubation, images were captured by using EVOS™ XL Core Imaging System (cat#AMEX1000).

## Mouse models of lung cancer

*Orthotopic model of lung cancer*: Lung tumors were generated in mice using Lewis Lung Carcinoma (LLC) cells as previously described (Doki et al, 1999). Briefly, log-phase cell cultures of LLC cells were collected and re-suspended at a cell density of 1 × 10$^4$ cells in 20 μL of 1:1 PBS: Matrigel. Six to eight weeks old mice (C57Bl/6 wild-type control, endFoxf1$^{+/-}$, and endFoxf1$^{OE}$) were anesthetized using isofluorane. A small incision was made on the left chest wall, subskin fat and muscles were separated from the

coastal bones and using a 0.3-mL insulin syringe, 10,000 mCherry-labeled LLC tumor cells were injected into the left lung through the intercostal space. The muscles were sutured using absorbable sutures and the skin incision was closed with a surgical skin clip. Mice were sacrificed on days 14, 21, or 28 post tumor cell inoculation. Lungs, and mediastinal lymph nodes were harvested, and tumor sizes were measured. Tumor volumes were calculated as ½ (L × W × W) where L is the largest tumor diameter and W is tumor diameter perpendicular to L. Tumor-bearing lungs were collected for histology or tumors were microdissected and CD45$^-$CD31$^+$ ECs were isolated by flow cytometry. RNA was extracted from sorted ECs for qRT-PCR or RNA-seq analysis. *Urethane-induced lung carcinogenesis*: Carcinogen-induced lung tumors were generated as previously described (Miller et al, 2003). Briefly, 6–8 weeks old mice were injected with urethane (25 mg; Sigma) intra-peritoneally once a week for 10 weeks and administered tamoxifen (3 mg; Sigma) orally once a month for 8 months. Mice were sacrificed 32 weeks after first urethane injection. Lungs were harvested, and numbers of visible tumor nodules were counted. Tumor nodules were classified into <1 mm, ~1 mm, and >1 mm diameter.

## Patient samples

De-identified non-small cell lung cancer (NSCLC) patient paraffin-imbedded lung sections or fresh tissue samples were obtained from the University of Cincinnati Biorepository (Appendix Table S1). Utilization of de-identified human samples was reviewed and approved by the Ethics Committee of the University of Cincinnati. A single-cell suspension was prepared from fresh lung cancer patient tissue samples using enzyme cocktail (dispase, and liberase), and endothelial cells (CD45$^-$CD31$^+$) were sorted using magnetic beads (Miltenyi Biotec). RNA was extracted from sorted endothelial cells and subjected to qRT-PCR analysis. Lung cancer patient survival was determined using online survival analysis software (Gyorffy et al, 2013).

## Histology and immunostaining

Lung paraffin sections were used for H&E staining, and paraffin sections or frozen sections were used for immunostaining as previously described (Cai et al, 2016). The list of antibodies used for immunostaining can be found in the Appendix Table S2. For

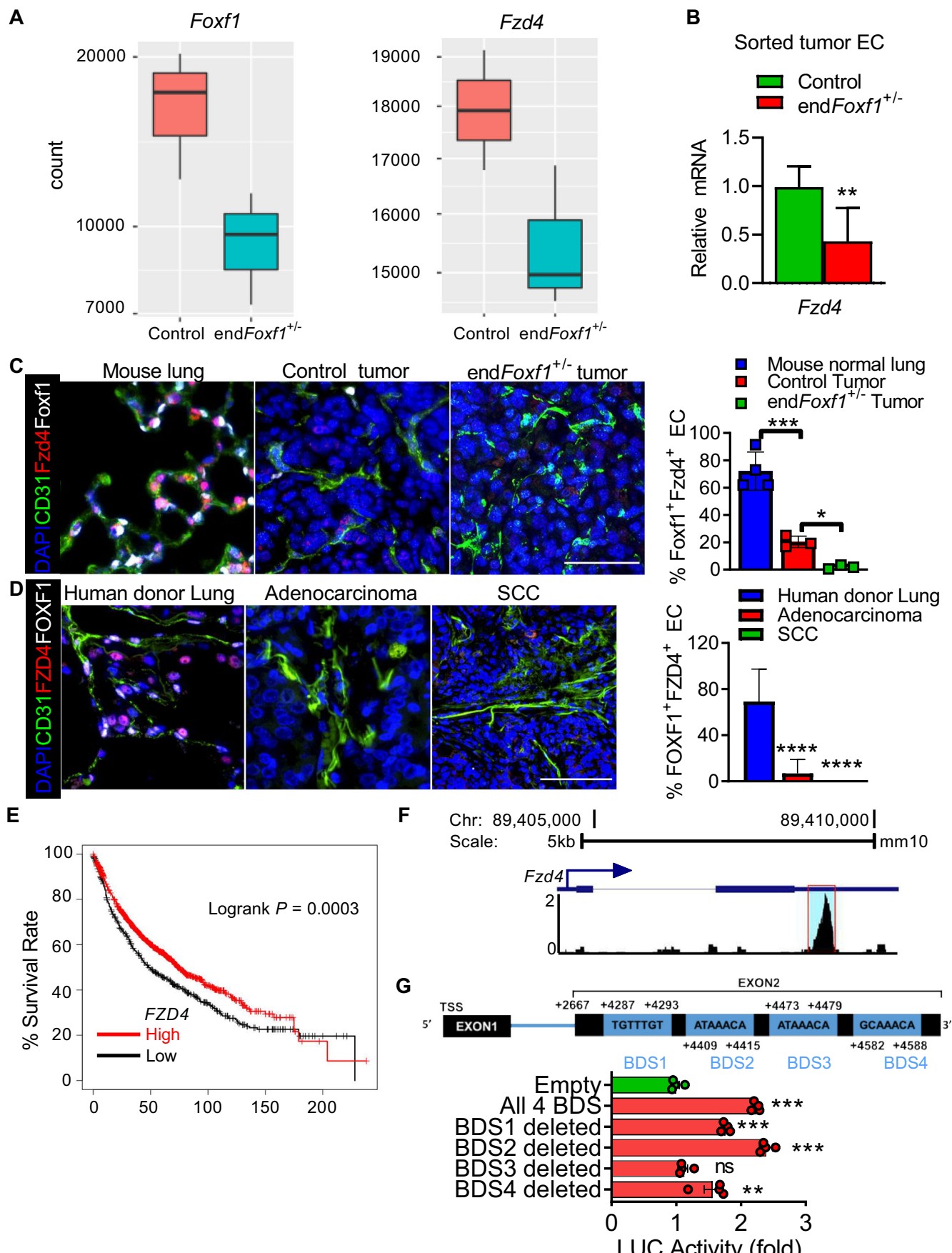

◀ **Figure 6. FZD4 is decreased after deletion of *Foxf1* in lung endothelial cells.**

(A) RNA-seq analysis shows decreased *Foxf1* and *Fzd4* mRNA transcripts in end*Foxf1*+/− TECs compared to controls. ($N = 3$ per group). Boxes show median, Q1 and Q3 quartiles and whiskers up to 1.5× interquartile range. (B) *Fzd4* mRNA is decreased in FACS-sorted end*Foxf1*+/− TECs compared to control TECs shown by qRT-PCR. β-actin mRNA was used for normalization. ($N = 8$–10 mice per group). **$P = 0.001$. (C) Immunostaining shows that Fzd4 (red) is expressed in CD31+ endothelial cells (green) in normal mouse lung (left panel). Deletion of *Foxf1* decreases Fzd4 in CD31+ endothelial cells within LLC tumors of end*Foxf1*+/− mice compared to controls. ($N = 3$–4 per group). The normal lung tissue image is the same as in Figure EV1B. Scale bar $= 50\,\mu$m. ***$P = 0.0002$; *$P = 0.0479$. (D) Co-localization studies show decreased FZD4 (red) staining in CD31− (green) endothelial cells within human lung adenocarcinomas and SCC compared to donor lungs. ($N = 7$–16 per group). Scale bar $= 50\,\mu$m. ****$P < 0.0001$ compared to donor lungs. (E) TCGA data mining show that lower *FZD4* levels predict poor overall survival in NSCLC patients. (F) ChIP-seq shows direct binding of FOXF1 protein to *Fzd4* promoter region in mouse MFLM-91U endothelial cells. (G) Schematic drawing of the pGL2-*Fzd4*-Luc construct with *Fzd4* DNA containing the FOXF1-binding sites (BDS) (top panel). In co-transfection experiments, CMV-FOXF1 expression vector increased transcriptional activity of the full-length *Fzd4* reporter compared to CMV-empty vector (bottom panel). Deletion of BDS1, 3 and 4, but not BDS2, diminished CMV-FOXF1-induced transcriptional activation of the *Fzd4* region. ($n = 4$ samples per group). ****$P < 0.0001$; ****$P = 0.0001$ compared to CMV-empty group. Data information: Data represent different numbers ($N$) of biological replicates. The data with error bars are shown as mean ± SEM. Statistical analysis was performed using the two-tailed unpaired-sample Student *t* test (B), logrank test (E), or one-way ANOVA followed by Tukey's post hoc test (C, D, G). Source data are available online for this figure.

immunofluorescence imaging, secondary antibodies conjugated with Alexa Fluor 488, Alexa Fluor 594, or Alexa Fluor 647 (Invitrogen/Molecular Probes) were used as described (Cai et al, 2016). Cell nuclei were counterstained with DAPI. Images were obtained using a Zeiss AxioPlan 2 microscope. *Tumor hypoxia:* Hypoxia within tumors was detected as previously described (Cantelmo et al, 2016). Briefly, tumor-bearing mice were injected with 60 mg/kg pimonidazole hydrochloride (hypoxyprobe kit) 1 h prior to sacrifice. To detect the formation of pimonidazole adducts, paraffin sections were immunostained with Hypoxyprobe-1-Mab1 following the manufacturer's instructions. *Tumor vessel perfusion:* A vascular perfusion assay was carried out as previously described (Cantelmo et al, 2016). Tumor-bearing mice were intravenously injected with 0.05 mg fluorescein-labeled *Lycopersicon esculentum* (Tomato) lectin (Vector Laboratories) 10 min prior to sacrifice. The perfused area was defined as lectin+ CD31+ area.

## RNA-seq and single-cell RNA-Seq data analysis

LLC tumors were microdissected from control and end*Foxf1*+/− mice 21 days post tumor cell injection. Endothelial cells (CD45−CD31+) cells were sorted from the microdissected lung tumors using FACS, and RNA was extracted and sequenced. Differentially expressed genes were identified using DESeq2, and both *P* value and fold change were calculated. For molecular pathway analysis, the functional annotation clustering was performed using DAVID Bioinformatics Resources Ver.6.8. The KEGG (Kyoto Encyclopedia of Genes and Genomes) pathway enrichment analysis was used to produce the annotation data. Functional clusters with Benjamini-corrected P value < 0.05 were considered statistically significant. The mouse scRNA-seq (ArrayExpress: E-MTAB-7458 (Goveia et al, 2020)) raw counts were retrieved from the website (https://www.vibcancer.be/software-tools/lungTumor_ECTax), and were analyzed using the Seurat R package as previously described (Wang et al, 2022; Wang et al, 2021). Total $n = 3841$ of normal lung EC (NEC) and $n = 10,012$ of tumor-associated EC (TEC) were included in the analysis.

## qRT-PCR

FACS-sorted lung endothelial cells were used for RNA extraction. cDNA was synthesized from RNA using iScript cDNA synthesis kit and qRT-PCR was carried out using Taqman probes listed in Appendix Table S3.

## Generation of the mouse *Fzd4* reported construct and luciferase assay

The mouse *Fzd4* DNA regulatory region 400 bp (DNA sequence +4261 to +4660 containing four FOXF1-binding site) was cloned into luciferase plasmid. The insert was placed before minimal promoter and luciferase plasmid. Fzd4 mutant constructs were generated using site-directed mutagenesis. A dual luciferase reporter assay was performed on Hek293T cells co-transfected with luciferase reporters and either a CMV-empty or CMV-*Foxf1* overexpression plasmid.

## Nanoparticle generation and delivery of plasmid

The PBAEs were synthesized using two-step Aza-Michael addition synthesis as described (Bian et al, 2023). To label the PBAE nanoparticle, the DyLight 650 NHS ester (ThermoFisher Scientific) was mixed with the nanoparticle at a mass ratio of 1:100. *Foxf1* plasmid was generated by cloning DNA (3HA-RFP-Foxf) into the Enhanced Episomal Vectors (EEV) empty plasmid vectors (SBI). The PBAE polymer (300 μg) was used to encapsulate 40 μg of plasmid DNA (CMV-*Fzd4* or CMV-empty). Nanoparticles (250 μl) were delivered to mice via tail vein or eye vein.

## RNAscope in situ hybridization assay

Assay was performed according to a protocol developed by Advanced Cell Diagnostics (ACD), using in situ probes designed by ACD, the RNAscope Multiplex Fluorescent Reagent Kit (v.2), and Opal dyes (Akoya Biosciences, 1:500 dilution for Opal 570 and 690 dyes, 1:1000 dilution for Opal 520 dyes). Nuclei were counterstained with DAPI, and tissue sections were mounted in Prolong Gold antifade reagent (Invitrogen). Proprietary (ACD) probes used: Mouse, Mm-Foxf1(473051), Mm-Fzd4-C3(404901-C3), Mm-Aplnr-C2(436171-C2), Mm-Cdh5(312531), Mm-Cldn5-C3(491611-C3). Tissue slides were photographed with a wide-field Nikon i90 or Nikon confocal microscope and quantified using the Nikon's NIS-Elements AR (Advanced Research, ver. 5) software.

## Statistical analysis

Statistical significance differences in measured variables between experimental and control groups were assessed by Student's *t* test

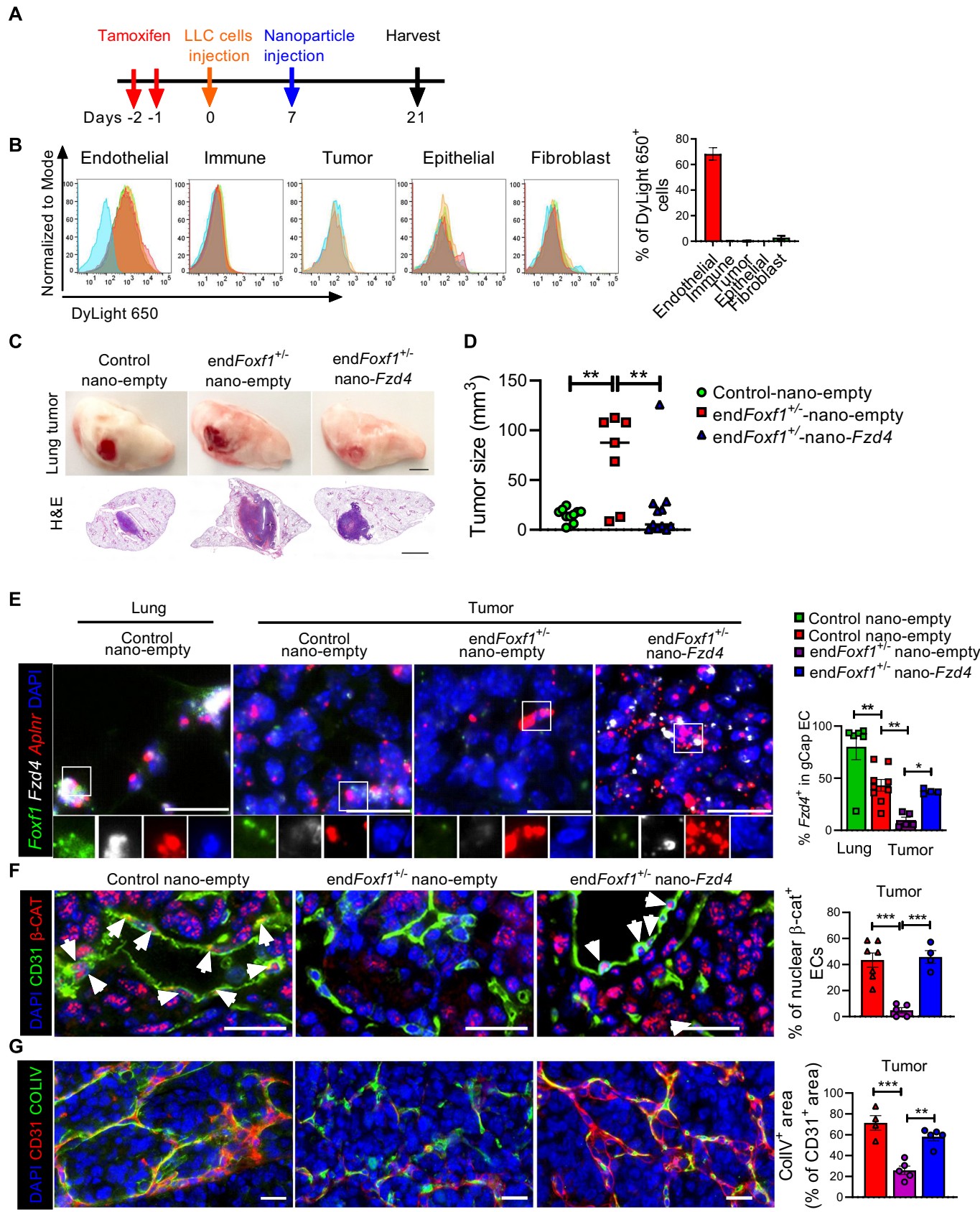

**Figure 7. FOXF1 regulates WNT signaling in endothelial cells through the FZD4 receptor.**

(A) Schematic diagram of nanoparticle delivery. Fluorescently labeled nanoparticles (DyLight 650$^+$) were delivered I.V. to tumor-bearing control and end$Foxf1+/−$ mice at day 7 after LLC tumor cells inoculation and the presence of nanoparticles was detected two weeks after nanoparticle treatment. (B) Using microdissected lung tumors from control mice, flow cytometry analysis shows the presence of PBAE nanoparticles in endothelial (CD31$^+$/CD45$^−$) cells, but not in hematopoietic/immune cells (CD45$^+$/CD31$^-$), epithelial cells (CD326$^+$) or fibroblasts (CD326$^−$/CD45$^−$/CD31$^-$). ($n = $3–4 mice per group). (C, D) Nanoparticle delivery of the $Fzd4$ vector (nano-Fzd4) to endothelial cells decreases lung tumor sizes in end$Foxf1+/-$ mice compared to treatment with nanoparticles containing CMV-empty vector (nano-Empty). ($N = $7–11 mice per group). Scale bar (top) $= 0.2$ cm. Scale bar (bottom) $= 900$ μm. (Control compared to end$Foxf1+/−$ nano-Empty: **$P = 0.0035$; end$Foxf1+/−$ nano-Empty compared to nano-Fzd4: **$P = 0.008$). (E) An increased number of $Fzd4$-positive endothelial cells in lungs of Nano-$Fzd4$ treated mice is shown using RNAscope ($N = 4$–9 mice per group). Endothelial cells were visualized using $Aplnr$ mRNA. Data presented as mean ± SEM. *$P < 0.05$. Scale bar $= 20$ μm. (Lung Control compared to Tumor Control: **$P = 0.0013$; Tumor Control compared to Tumor end$Foxf1+/−$ nano-Empty: **$P = 0.0044$; Tumor end$Foxf1+/−$ nano-Empty compared to nano-Fzd4: *$P = 0.0394$). (F) Co-localization studies show increased nuclear β-CATENIN (red) staining in CD31$^+$ (green) endothelial cells within LLC tumors of end$Foxf1^{+/−}$ mice treated with nano-$Fzd4$ compared to nano-Empty groups. (Arrows: double-positive cells; $N = 4$–7 mice per group). Scale bar $= 20$ μm. Tumor Control compared to Tumor end$Foxf1+/−$ nano-Empty: ***$P = 0.0002$; Tumor end$Foxf1+/−$ nano-Empty compared to nano-Fzd4: ***$P = 0.0003$). (G) Treatment of end$Foxf1^{+/−}$ mice with nano-$Fzd4$ increases the basement membrane coverage in tumor vessels, shown by increased Collagen IV immunostaining (red) and increased percentage of ColIV+ area relative to CD31+ area (green). The percentage of positive area was counted in five random fields and presented as mean ± SEM. ($N = 4$–5 mice per group). Scale bar $= 20$ μm. Tumor Control compared to Tumor end$Foxf1+/−$ nano-Empty: ***$P = 0.0002$; Tumor end$Foxf1+/−$ nano-Empty compared to nano-Fzd4: **$P = 0.0012$). Data information: Data represent different numbers ($N$) of biological replicates. The data with error bars are shown as mean ± SEM. Statistical analysis was performed using the one-way ANOVA followed by Tukey's post hoc test (D–G). Source data are available online for this figure.

## The paper explained

### Problem

Tumor-associated angiogenesis is essential to promote tumor growth and metastases. Tumor blood vessels are abnormal, endothelial layer is disorganized and leaky. Lung has a robust capillary network, providing unique microenvironment for highly metastatic non-small cell lung cancers (NSCLC). Presence of tumor cells induce reprogramming of normal lung endothelial cells (EC) into tumor-associated endothelial cells (TEC), which promote tumor progression and metastasis. The transcriptional regulators that control EC-to-TEC transition are poorly understood.

### Results

We have identified the Forkhead Box transcription factor 1 (FOXF1) as a critical regulator of EC-to-TEC transition. FOXF1 is expressed in normal EC, but rapidly decreases in TEC of mouse and human NSCLC. Low FOXF1 predicts poor survival in NSCLC patients. While endothelial-specific deletion of FOXF1 in mice destabilizes tumor vessels and promotes lung cancer growth and metastasis, the endothelial over-expression of FOXF1 normalizes tumor vessels and inhibits lung tumor progression. FOXF1 deficiency decreases Wnt/β-catenin signaling in TECs through direct transcriptional activation of $Fzd4$. Nanoparticle delivery of $Fzd4$ cDNA to endothelial cells rescued Wnt/β-catenin signaling in TECs, normalized tumor vessels and inhibited the progression of lung cancer.

### Impact

Our study identifies the novel molecular mechanisms used by lung endothelial cells to regulate tumorigenesis which involves FOXF1/FZD4/WNT signaling. We have demonstrated that maintaining FOXF1/FZD4 signaling in pulmonary EC via genetic or gene therapy is beneficial to normalize tumor vessels and inhibit lung cancer progression. Nanoparticle delivery of FZD4 cDNA represents a promising therapeutic strategy to treat NSCLC.

independently with similar results more than three times. All experiments were performed randomization. All animal experiments and in vitro studies were performed and analyzed in a blinded manner. All samples were used for analysis and there were no inclusion/exclusion criteria.

## Ethical statement

The Institutional Review Board of the Cincinnati Children's Hospital Medical Center (Federalwide Assurance #00002988) approved all the studies with human tissue samples (IRB protocol #2017-4321). Human lung tissue specimens were obtained from tissue repository at the University of Cincinnati Medical Center that provides de-identified human biospecimen procurement and banking services in support of basic, translational, and clinical research. All patients signed informed consent prior to tissues collection. All animal studies were approved by the Cincinnati Children's Research Foundation Institutional Animal Care and Use Committee and covered under our animal protocol (IACUC2016-0070). The Cincinnati Children's Research Foundation Institutional Animal Care and Use Committee is an AAALAC and NIH-accredited institution (NIH Insurance #8310801). All mice were kept under SPF (specific-pathogen-free) conditions in 12/12 light/dark cycle, 18–23 °C, and 40–60% humidity. Both males and females were used for studies. The experiments conformed to the principles set out in the WMA Declaration of Helsinki and the Department of Health Services Belmont Report.

## Graphics

Synopsis image graphics were created with BioRender.com.

## Data availability

The RNA-seq data uploaded to GEO database can be accessed at GSE255969.

(two-tailed) or Mann–Whitney test. Multiple groups were compared using one-way analysis of variance (ANOVA) followed by Tukey's post hoc test. $P$ values < 0.05 were considered significant. Values for all measurements were expressed as the mean ± standard deviation or as the mean ± standard error of the mean. Statistical analysis was performed, and data were graphically displayed using GraphPad Prism v.9.0 for Windows (GraphPad Software, Inc., San Diego, CA). All experiments in this study have been repeated

# Peer review information

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

## Acknowledgements

This work was supported by the NIH grants R01 HL132849 (TVK), R01 HL158659 (TVK), R01 HL141174 (VVK), R01 HL149631 (VVK), R01 HL152973 (VVK/TVK).

## Author contributions

**Fenghua Bian**: Formal analysis; Validation; Investigation; Visualization; Methodology. **Chinmayee Goda**: Conceptualization; Formal analysis; Validation; Investigation; Visualization; Methodology; Writing—original draft. **Guolun Wang**: Formal analysis; Validation; Investigation; Visualization. **Ying-Wei Lan**: Formal analysis; Investigation; Visualization. **Zicheng Deng**: Formal analysis; Investigation; Visualization. **Wen Gao**: Formal analysis; Investigation; Visualization. **Anusha Acharya**: Formal analysis; Visualization; Methodology. **Abid A Reza**: Formal analysis; Investigation; Visualization. **Jose Gomez-Arroyo**: Formal analysis; Investigation; Visualization. **Nawal Merjaneh**: Investigation; Writing—review and editing. **Xiaomeng Ren**: Formal analysis; Investigation. **Jermaine Goveia**: Resources; Data curation; Software. **Peter Carmeliet**: Resources; Data curation; Software. **Vladimir V Kalinichenko**: Resources; Supervision; Project administration; Writing—review and editing. **Tanya V Kalin**: Conceptualization; Supervision; Validation; Visualization; Writing—original draft; Project administration; Writing—review and editing.

## Disclosure and competing interests statement

The authors declare no competing interests. T Kalin is a member of the EMM Editorial Board. This has no bearing on the editorial consideration of this article for publication.

# Expanded View Figures

**A**

**TCGA Lung Adenocarcinama patient**

Lower *FOXF1* mRNA quartile
n=294, median survival=50.00

Upper *FOXF1* mRNA quartile
n=292, median survival=undefined

*p*<0.0001
Hazard Ratio (logrank)=1.947 (1.530 to 2.477)

|  | Log-rank (Mantel-Cox) Test | Gehan-Breslow-Wilcoxon Test |
|---|---|---|
| Chi square | 28.01 | 25.15 |
| df | 1 | 1 |
| P value | <0.0001 | <0.0001 |
| P value summary | **** | **** |
| Are the survival curves sig different? | Yes | Yes |

**TCGA Lung Squamous Carcinoma patient**

Lower *FOXF1* mRNA quartile
n=195, median survival=40.00

Upper *FOXF1* mRNA quartile
n=195, median survival=70.30

*p*=0.0160
Hazard Ratio (logrank)=1.401 (1.065 to 1.843)

|  | Log-rank (Mantel-Cox) Test | Gehan-Breslow-Wilcoxon Test |
|---|---|---|
| Chi square | 5.805 | 4.235 |
| df | 1 | 1 |
| P value | 0.0160 | 0.0396 |
| P value summary | * | * |
| Are the survival curves sig different? | Yes | Yes |

**B**

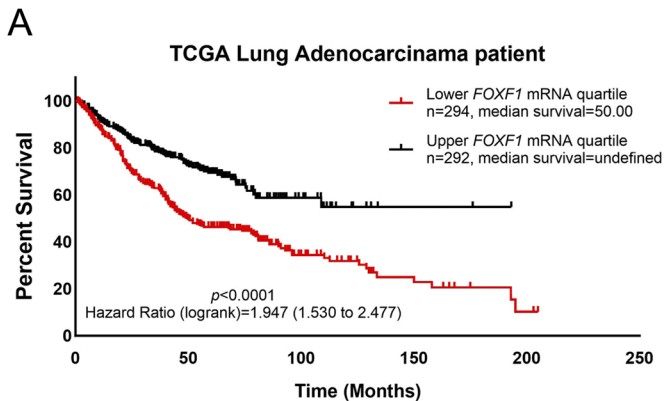

**C**

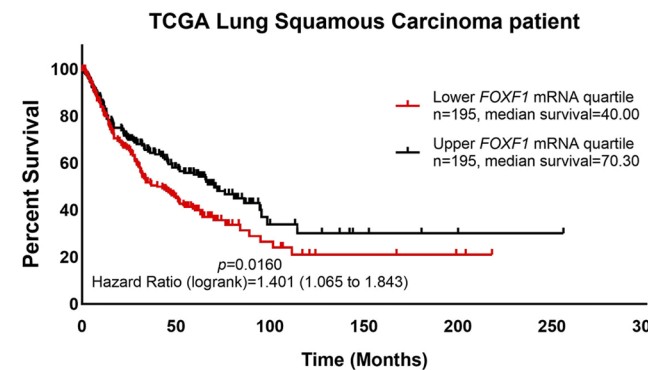

**D**

◄ **Figure EV1.  *FOXF1* expression is decreased in TECs of mouse LLC lung tumors.**

(**A**) TCGA data mining show that lower *FOXF1* mRNA levels in tumors predicts poor overall survival in patients with adenocarcinoma (AD) and squamous cell carcinoma (SCC). The lower and upper quartile for *FOXF1* expression in AD and SCC were compared. ($N = 1926$). (**B**) Co-localization studies demonstrate decreased FOXF1 protein (red) in CD31$^+$ endothelial cells (green) within mouse LLC tumors compared to normal lungs. The normal lung tissue image is the same as in Fig. 6C. ($N = 3$ mice per group). Scale bar = 100 μm. (**C**) Endothelial cells from normal lung (NEC) and LLC tumors (TEC) were visualized using uniform manifold approximation and projection (UMAP) after samples integration with Harmony. (**D**) Both *Foxf1* mRNA levels and the total number of FoxF1-expressing EC were decreased in LLC lung tumors compared to normal lung. Based on endothelial cell sub-clustering, decreased Foxf1 mRNA was detected in tumor-associated capillary, arterial and venous ECs. Foxf1 was not expressed in lymphatics. Colored by group; red- Donor EC, (NEC; $n = 3841$ cells); blue- EC from LLC tumors, (TEC, $n = 10012$ cells). Boxes show median, Q1 and Q3 quartiles and whiskers up to 1.5× interquartile range. Data information: Data represent different numbers ($N$) of biological replicates. The data with error bars are shown as mean ± SEM. *$P < 0.05$, ns: not significant, as determined using the logrank test (**A**), or Wilcoxon rank-sum test (**D**).

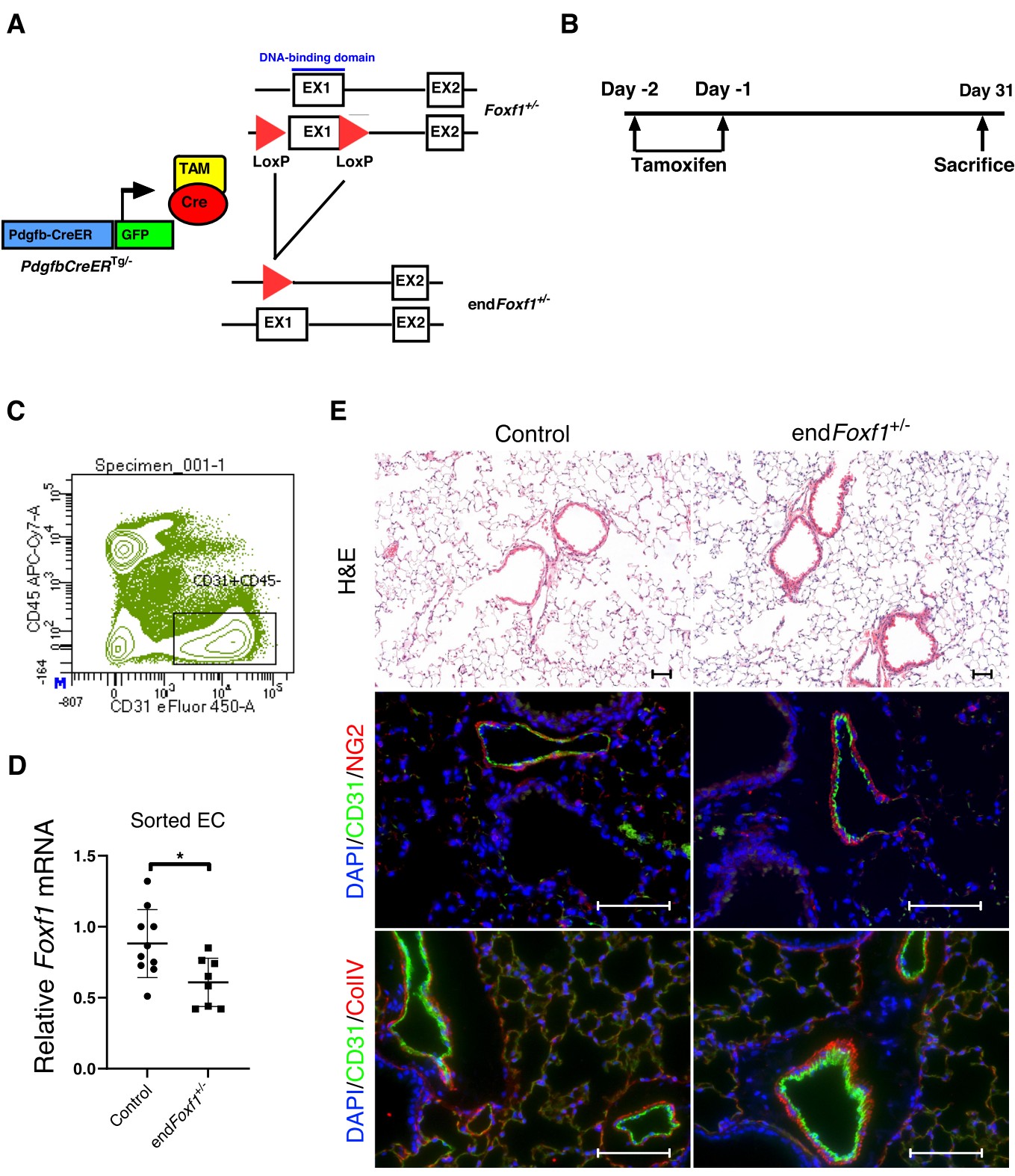

◀ **Figure EV2. Conditional deletion of one *Foxf1* allele from endothelial cells does not affect lung histology.**

(A) Schematic representation of breeding strategy for the conditional deletion of one allele of *Foxf1* from endothelial cells (end*Foxf1*$^{+/-}$ mice). *Foxf1*$^{F/F}$ mice were crossed with *Pdgfb-cre*ER$^{+/-}$ mice to generate *Pdgfb-cre*ER$^{+/-}$; *Foxf1*$^{F/+}$ mice. *Foxf1*$^{F/+}$ littermates were used as controls. (B) Schematic representation of tamoxifen treatment to delete *Foxf1* from endothelial cells. (C) The gating strategy for FACS-sorting. Single, live cells from the lungs were gated to identify endothelial cells (CD45$^-$ CD31$^+$). (D) qRT-PCR shows decreased *Foxf1* mRNA in endothelial cells isolated from end*Foxf1*$^{+/-}$ lungs compared to control lungs. *Actb* mRNA was used for normalization. (N = 8–10 mice per group). *$P$ = 0.0151. (E) H&E staining (upper panels) shows no morphological differences between tamoxifen-treated control *Foxf1*$^{F/+}$ lungs and end*Foxf1*$^{+/-}$ lungs at 31 days after tamoxifen administration. Scale bar = 25 μm. No difference in pericyte coverage of blood vessels was found in control and end*Foxf1*$^{+/-}$ lungs is shown by co-localization of CD31 (green) with NG2 (red) (Middle panels). Blood vessels in end*Foxf1*$^{+/-}$ and control lungs have similar basement membrane as shown by co-localization of CD31 (green) with Collagen IV (red). (Bottom panels). Scale bar = 100 μm. Data information: Data represent different numbers (N) of biological replicates. The data with error bars are shown as mean ± SD. Statistical analysis was performed using Mann–Whitney Two-tailed test (D). Source data are available online for this figure.

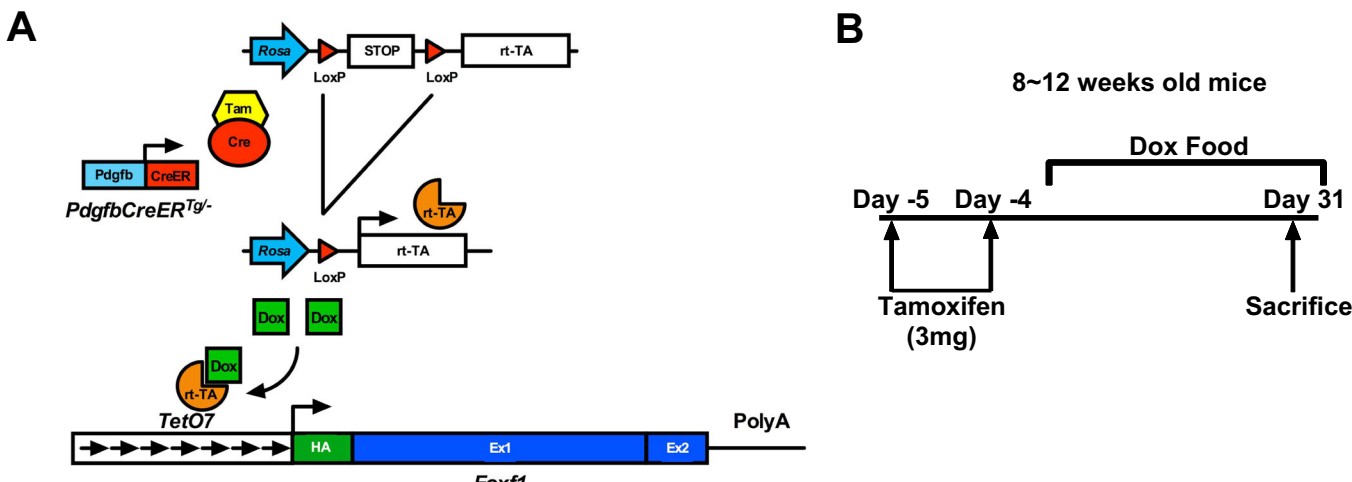

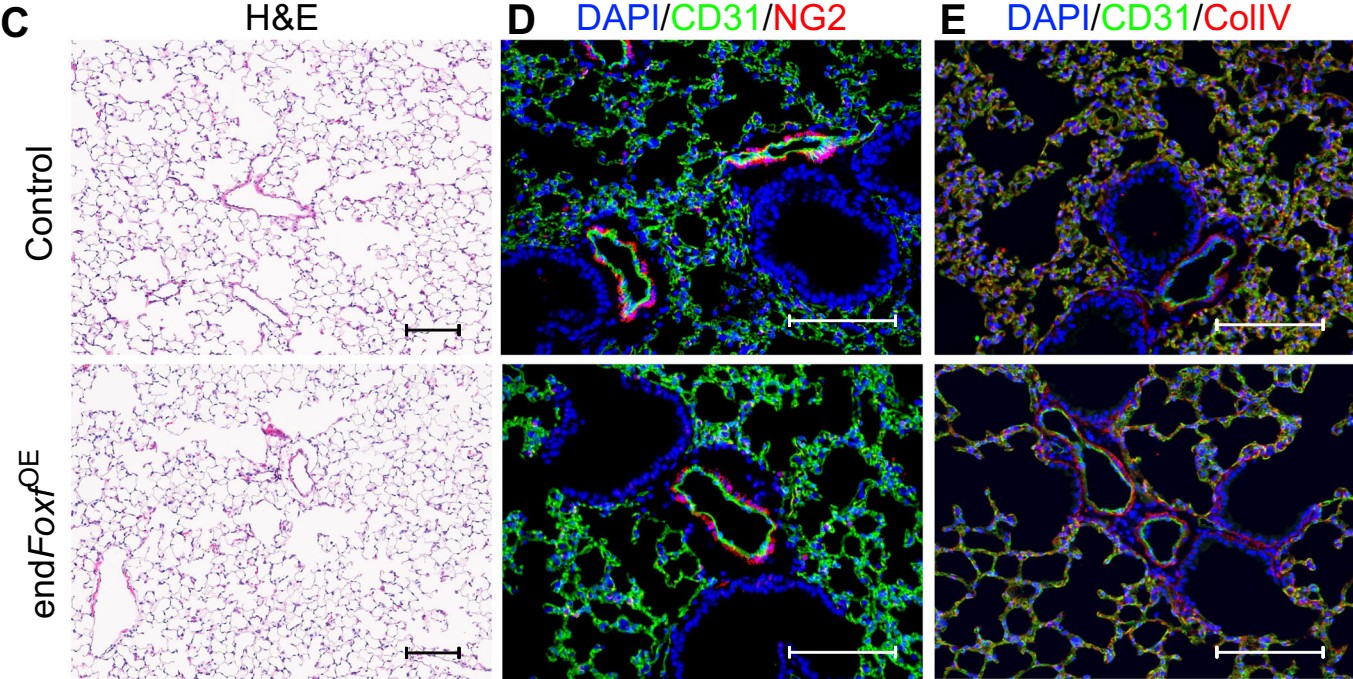

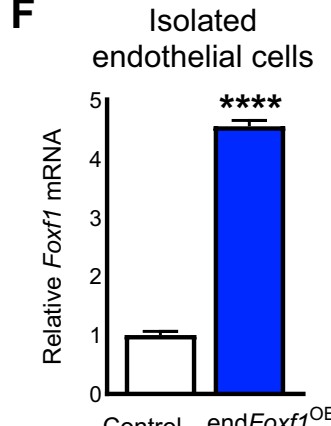

**Figure EV3. Overexpression of FOXF1 in endothelial cells does not affect lung architecture.**

(A) Schematic representation shows breeding strategy for overexpression of FOXF1 in endothelial cells (end*Foxf1*^OE). (B) Schematic representation of tamoxifen and doxycycline treatment to overexpress FOXF1 in endothelial cells. (C) H&E staining shows no morphological differences between tamoxifen+doxycycline-treated control and end*Foxf1*^OE mouse lungs at 31 days after the treatment. Scale bar = 50 μm. (D) No difference in pericyte coverage of blood vessels in control and end*Foxf1*^OE lungs is shown by co-localization of CD31 (green) with NG2 (red). (E) Blood vessels in end*Foxf1*^OE and control lungs have similar basement membrane as shown by co-localization of CD31 (green) with Collagen IV (red). (F) *Foxf1* mRNA is increased by ~4.5-fold in CD45$^-$CD31$^+$ endothelial cells FACS-sorted from tamoxifen+doxycycline-treated end*Foxf1*^OE mice compared to controls. β-*actin* mRNA was used for normalization. ($N = 3$ mice per group). Scale bar = 100 μm. ****$P < 0.0001$. Data information: Data represent different numbers ($N$) of biological replicates. The data with error bars are shown as mean ± SEM. Statistical analysis was performed using the two-tailed unpaired-sample Student *t* test (F). Source data are available online for this figure.

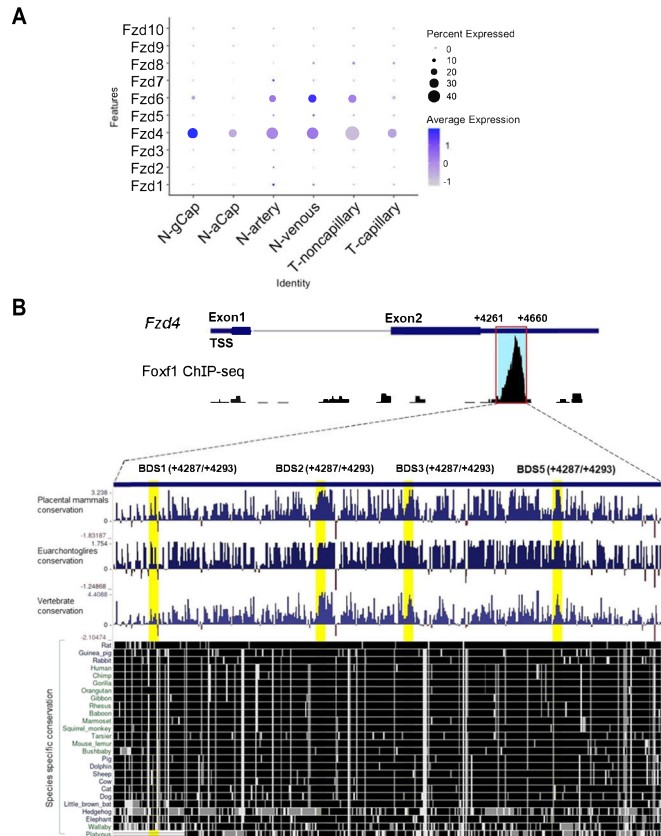

**Figure EV4. Fzd4 is a direct transcriptional target of FOXF1.**

(A) Expression of Frizzled receptor mRNAs in lung endothelial cells based on scRNA-seq analysis. (B) Analysis of ChIP-seq dataset from MFLM-91U endothelial cells shows FOXF1-binding picks in Fzd4 gene region located 3′ to the exon 2 of the Fzd4 coding sequence (top panel). Homology of the Fzd4 gene with FOXF1-binding sites (BDS) is shown for several mammalian species (bottom panel).

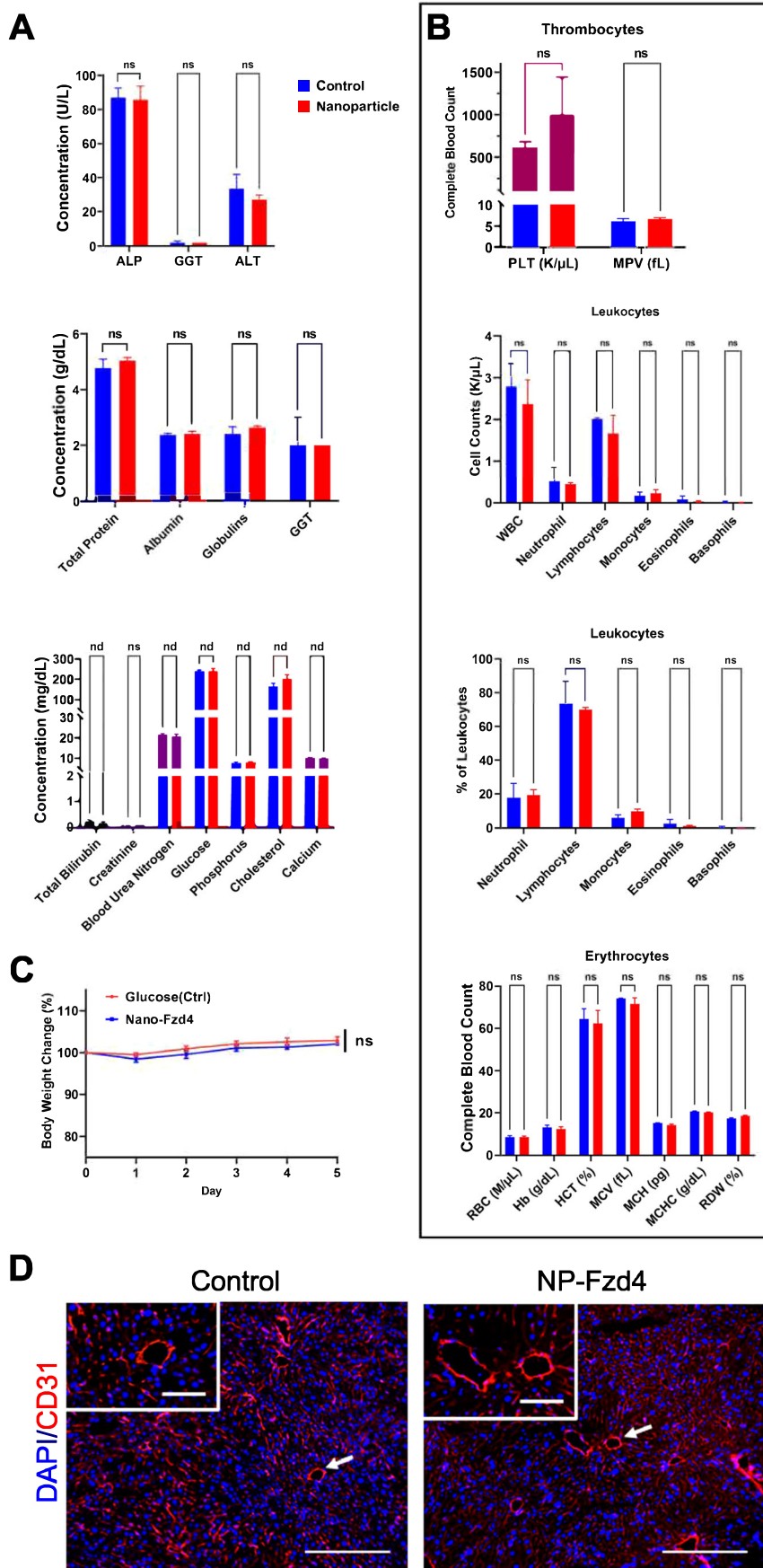

◀ **Figure EV5. Treatment with intravenously injected PBAE nanoparticles carrying CMV- *Fzd4* (nano-Fzd4) is not toxic.**

(A) Normal liver functions: No differences were found in the concentrations of total protein, Albumin, Globulins, ALP, Total Bilirubin, GGT, and ALT in peripheral blood serum of mice treated with nano-Fzd4 compared to control mice treated with glucose. $N = 5$ mice per group. (B) Normal kidney functions: No differences in BUN and Creatinine levels were found in the nano-Fzd4 treated group compared to the control group. $N = 5$ mice per group. (C) No changes in hematologic parameters compared to the control group. $N = 5$ mice per group. (D) Nano-Fzd4 treatment does not change the histological appearance of endothelial cells in the liver shown with immunostaining for Pecam1 (CD31). Arrows indicated blood vessels. $N = 2$–3 mice per group. Data information: Data represent different numbers ($N$) of biological replicates. The data with error bars are shown as mean ± SEM. ns: not significant, as determined using two-tailed unpaired-sample Student $t$ test (**A**–**C**). Source data are available online for this figure.

