## [Peer Review File · EMBO Molecular Medicine]

FOXF1 Promotes Tumor Vessel Normalization and Prevents Lung Cancer Progression through FZD4

Fenghua Bian, Chinmayee Goda, Guolun Wang, Ying-Wei Lan, Zicheng Deng, Wen Gao, Anusha Acharya, Abid Reza, Jose Gomez-Arroyo, Nawal Merjaneh, Xiaomeng Ren, Jermaine Goveia, Peter Carmeliet, Vladimir Kalinichenko, and Tanya Kalin

Corresponding author: Tanya Kalin (tatianakalin@arizona.edu)

Review Timeline:

Submission Date:	24th Jul 23
Editorial Decision:	11th Aug 23
Revision Received:	19th Feb 24
Editorial Decision:	6th Mar 24
Revision Received:	19th Mar 24
Accepted:	21st Mar 24

Editor: Lise Roth

Transaction Report:

11th Aug 2023

Dear Tanya,

Thank you for the submission of your manuscript to EMBO Molecular Medicine. We have now received feedback from the three reviewers who agreed to evaluate your manuscript. As you will see from the reports below, the referees acknowledge the interest of the study and are overall supporting publication of your work pending appropriate revisions.

Addressing the reviewers' concerns in full will be necessary for further considering the manuscript in our journal, however, upon further consultation with the referees and discussion with my colleagues, we agreed that point 1. from referee #1 might be further reaching in the context of the current study, and we would suggest to discuss this point only. EMBO Molecular Medicine encourages a single round of revision and therefore, acceptance or rejection of the manuscript will depend on the completeness of your responses included in the next, final version of the manuscript. For this reason, and to save you from any frustrations in the end, I would strongly advise against returning an incomplete revision.

We are expecting your revised manuscript within three months, if you anticipate any delay, please contact us.

We require:

4) A .docx formatted letter INCLUDING the reviewers' reports and your detailed point-by-point responses to their comments. As part of the EMBO Press transparent editorial process, the point-by-point response is part of the Review Process File (RPF), which will be published alongside your paper.

5) A complete author checklist, which you can download from our author guidelines (<https://www.embopress.org/page/journal/17574684/authorguide#submissionofrevisions>). Please insert information in the checklist that is also reflected in the manuscript. The completed author checklist will also be part of the RPF.

6) Please note that all corresponding authors are required to supply an ORCID ID for their name upon submission of a revised manuscript.

7) It is mandatory to include a 'Data Availability' section after the Materials and Methods. Before submitting your revision, primary datasets produced in this study need to be deposited in an appropriate public database, and the accession numbers and database listed under 'Data Availability'. Please remember to provide a reviewer password if the datasets are not yet public (see <https://www.embopress.org/page/journal/17574684/authorguide#dataavailability>).

8) For data quantification: please specify the name of the statistical test used to generate error bars and P values, the number (n) of independent experiments (specify technical or biological replicates) underlying each data point and the test used to calculate p-values in each figure legend. The figure legends should contain a basic description of n, P and the test applied. Graphs must include a description of the bars and the error bars (s.d., s.e.m.). Please provide exact p values.

10) We replaced Supplementary Information with Expanded View (EV) Figures and Tables that are collapsible/expandable online. A maximum of 5 EV Figures can be typeset. EV Figures should be cited as "Figure EV1, Figure EV2" etc... in the text and their respective legends should be included in the main text after the legends of regular figures.

13) Author contributions: CRediT has replaced the traditional author contributions section because it offers a systematic machine readable author contributions format that allows for more effective research assessment. Please remove the Authors Contributions from the manuscript and use the free text boxes beneath each contributing author's name in our system to add specific details on the author's contribution. More information is available in our guide to authors.

16) As part of the EMBO Publications transparent editorial process initiative (see our Editorial at <http://embomolmed.embopress.org/content/2/9/329>), EMBO Molecular Medicine will publish online a Review Process File (RPF) to accompany accepted manuscripts.

In the event of acceptance, this file will be published in conjunction with your paper and will include the anonymous referee reports, your point-by-point response and all pertinent correspondence relating to the manuscript. Let us know whether you agree with the publication of the RPF and as here, if you want to remove or not any figures from it prior to publication. Please note that the Authors checklist will be published at the end of the RPF.

I look forward to receiving your revised manuscript.

With kind regards,

Lise

***** Reviewer's comments *****

Referee #1 (Remarks for Author):

The manuscript provides valuable insights into the critical role of Forkhead box F1 (FOXF1) in reprogramming tumor-associated endothelial cells (TECs) within non-small cell lung cancers (NSCLC). The study highlights FOXF1 as a pivotal transcriptional regulator orchestrating the endothelial cell (EC)-to-TEC transition, supported by compelling evidence of its reduced expression in TECs from both human and mouse NSCLC tumors. Notably, the observed low levels of FOXF1 in NSCLC correlate with poor overall patient survival. The authors effectively utilize genetically modified mice and conduct comprehensive molecular analyses, demonstrating that endothelial-specific deletion of FOXF1 fosters lung tumor growth and metastasis by inducing structural and functional abnormalities in tumor vasculature. Conversely, endothelial-specific overexpression of FOXF1 normalizes tumor vessels and inhibits lung cancer progression. A significant implication arises from their discovery of decreased Wnt/ β -catenin signaling in TECs due to FOXF1 deficiency, which stems from direct transcriptional activation of Fzd4. To our interest, the authors show that restoring FZD4 expression through endothelial-specific nanoparticle delivery of Fzd4 cDNA rescues Wnt/ β -catenin signaling, normalizes tumor vessels, and effectively inhibits lung cancer progression.

In conclusion, the study presents interesting findings regarding the role of FOXF1 in TEC reprogramming within NSCLC. However, there are important issues that need to be addressed, particularly with respect to the novelty of the findings, the relevance of the downstream effector pathway, the nanoparticle delivery's potential impact, and the absence of human models, before it can be deemed suitable for publication in the EMBO Molecular Medicine.

Major concern:

1. Previous studies have investigated the angiogenic role of FOXF1 in other disease models or experimental models, as evidenced by publications from both the authors and other researchers (Bian et al., 2023, Nature Comms 14:2560; Sturtzel et al., Front. Bioeng. Biotechnol., 2018 Jun 14;6:76). It would be valuable for the authors to elucidate what factors or mechanisms contribute to the divergent roles of FOXF1 in different diseases, as this understanding could provide critical insights into its context-dependent functions and potential implications for therapeutic targeting in specific conditions.
2. The downstream effector pathway of FOXF1 identified in this study, Wnt/ β -catenin signaling, is well-known for its significant role in regulating cancer cell growth and progression (Nusse et al., Cell 2017 Jun 1;169(6):985-999). Moreover, there is existing evidence suggesting that FOXF1 can also regulate Wnt/ β -catenin signaling in other cell types (Shen et al., Ebiomedicine 2020:102626). These findings raise concerns about the novelty of their current findings and the potential efficacy of solely targeting the endothelial-FOXF1-FZD4-Wnt/ β -catenin axis to effectively repress tumor growth. To provide a comprehensive analysis, the authors should address this crucial aspect in their discussion, considering the multifaceted roles of Wnt/ β -catenin signaling and FOXF1 in cancer biology.
3. The utilization of nanoparticles for delivering Fzd4 cDNA raises valid concerns, considering the known toxicity associated with nanoparticles in human trials (Yang et al., Annual Review of Pharmacology and Toxicology, 61, p269-289). Thus, the authors should address potential safety issues by investigating whether the administration of these nanoparticles leads to any pathological damage in the mice model used in the study. Additionally, given that the angiogenic role of Fzd4 is less characterized, it is crucial to assess whether these nanoparticles exert any adverse effects on normal blood vessel endothelial cells, particularly in the liver, as nanoparticles are known to accumulate in this organ.
4. The absence of human models and the use of a single mouse lung cell line in this study present significant challenges when

attempting to translate the authors' findings into clinical applications. To address this limitation, the authors could enhance the relevance of their study by conducting additional experiments using human umbilical vein endothelial cells (HUVECs) or tumor-derived endothelial cells (TECs). Tube formation assays or in vivo matrix plug assays with these human cells would allow investigation of FOXF1's angiogenic role under conditions that better mimic the human microenvironment. Furthermore, to gain insights into the relevance of their findings in an in vivo setting closer to human disease, the authors can perform co-injection experiments of human cancer cells and genetically modified TECs in nude mice, allowing evaluation of the impact on subcutaneous lung tumor growth.

Minor concerns:

1. To enhance the clinical significance of their findings, the authors should incorporate survival data from their own patient cohort in addition to utilizing the online TCGA database (Figure 1E).
2. To strengthen their findings in Figure 2C, the authors should provide representative images of metastatic section stained with H&E.
3. The absence of scale bars in Figure 5A and B is a notable oversight that should be addressed. Furthermore, in Figure 5A and B, the presentation of a tumor adhering to the lung surface, rather than growing inside the lung, appears unusual and warrants clarification from the authors.
4. Once again, it is essential for the authors to include representative images of H&E-stained metastatic sections to substantiate their findings in Figure 5D.
5. To ensure transparency and reproducibility, the authors should include the "n" number, representing the total number of cells analyzed, for their single-cell sequencing results. This crucial information is currently missing in both the methods section and figure legend.
6. The authors are advised to thoroughly review the manuscript for spelling and grammar errors. In particular, there is a spelling mistake in Figure 6E, where "survial" should be corrected to "survival."

Referee #2 (Remarks for Author):

Bian, Goda, Wang and colleagues report on the role of FOXF1 in lung endothelial cells in non-small cell lung cancer. Decreased expression of FOXF1 in tumor vessels altered vessel permeability and promoted cancer progression. Mechanistically, FOXF1 activated expression of Fzd4 and restoring its expression (using nanoparticle delivery of cDNA) was also sufficient to normalize tumor vessels. Overall, the manuscript is interesting, clearly written and describes important results.

Comments:

- It would be interesting to know if the survival analysis presented in Fig 1D holds true in both adenocarcinoma and squamous cell carcinoma.
- Why is SOX17 preferred to CD31 to identify endothelial cells in mouse tissues in Fig 1 but not later in Fig 3? Do these markers identify the same cells in mice?
- Fig 4C shows dual b-catenin/CD31 positivity, rather than nuclear b-catenin as is suggested in the results section text. Likewise, it is not clearly shown that nuclear b-catenin is reduced in the human data presented in Fig 4E-G. In Fig 5I and Fig 7F nuclear b-catenin is quantified but the same y-axis label is used as in previous figures that look at overall b-catenin expression. Ideally both quantifications would be shown for all experiments but this should be clarified.
- Did FOXF1 overexpression increase the expression of the other Wnt target genes identified by RNA seq analyses in Fig 4, or only Axin2 (Fig 5F)?
- The survival plots in Fig 1D and Fig 6E could be edited to appear similar (e.g. high expression in consistent color).

Typos:

- "we next used [a] publicly available scRNAseq dataset"
- "Cre-mediated recombination of [a] floxed stop codon"
- Fig 6E: "survi[v]al"
- "whether FOXF1 directly regulates transcription of [the] Fzd4 gene"
- "[A] previously published ChIP-seq dataset"

Referee #3 (Comments on Novelty/Model System for Author):

Combination of patient data with orthotopic mouse model of NSCLC with lung EC-specific FOX-F1 overexpression or deficiency are powerful tools to directly address the research question. Use of nanoparticles for EC-specific induction of FOX-F1 expression may have a high medical impact on lung cancer therapy

Referee #3 (Remarks for Author):

This is a very interesting study addressing a role of FoxF1 activity in EC-TEC endothelial phenotypic transition observed in lung cancer. The authors demonstrate that FoxF1 deficiency is associated with higher mortality in patients with non-small cell lung cancers (NSCLC). They also observed similar findings in mice with EC-specific FOX-F1 deficiency, but tumor development was

suppressed by FOX-F1 overexpression or gene transduction using EC-specific nanoparticles. The conclusions are supported by data, the rigor of study is strong, and impact high given the prospective of future use of EC-specific gene delivery to reduce mortality of patients with NSCLC. This reviewer has only a few minor comments to better clarify the mechanism and functional effects of EC-TEC endothelial phenotypic transition.

Minor comments:

1. Was increased CD31 expression associated with increased EC proliferation? This scenario might be consistent with decreased capillary perfusion in FoxF1 deficient lungs.
2. Given increased EC permeability in FoxF1 deficient lungs, were there any changes in VE-cadherin and claudin-5 levels observed? If so, were they controlled by same FZD4 mechanism?
3. Besides EC-TEC transformation, did the authors observe increased angiogenesis in lung tumors? If so, how can this effect be reconciliated with inhibited capillary angiogenesis in FoxF1-deficient embryos?

We would like to thank the Editor and the Reviewers for their time, efforts, and insightful comments. We greatly appreciate the constructive feedback and feel that the manuscript will be significantly improved by addressing the included comments.

***** Reviewer's comments *****

Referee #1 (Remarks for Author):

The manuscript provides valuable insights into the critical role of Forkhead box F1 (FOXF1) in reprogramming tumor-associated endothelial cells (TECs) within non-small cell lung cancers (NSCLC). The study highlights FOXF1 as a pivotal transcriptional regulator orchestrating the endothelial cell (EC)-to-TEC transition, supported by compelling evidence of its reduced expression in TECs from both human and mouse NSCLC tumors. Notably, the observed low levels of FOXF1 in NSCLC correlate with poor overall patient survival. The authors effectively utilize genetically modified mice and conduct comprehensive molecular analyses, demonstrating that endothelial-specific deletion of FOXF1 fosters lung tumor growth and metastasis by inducing structural and functional abnormalities in tumor vasculature. Conversely, endothelial-specific overexpression of FOXF1 normalizes tumor vessels and inhibits lung cancer progression. A significant implication arises from their discovery of decreased Wnt/ β -catenin signaling in TECs due to FOXF1 deficiency, which stems from direct transcriptional activation of Fzd4. To our interest, the authors show that restoring FZD4 expression through endothelial-specific nanoparticle delivery of Fzd4 cDNA rescues Wnt/ β -catenin signaling, normalizes tumor vessels, and effectively inhibits lung cancer progression.

In conclusion, the study presents interesting findings regarding the role of FOXF1 in TEC reprogramming within NSCLC. However, there are important issues that need to be addressed, particularly with respect to the novelty of the findings, the relevance of the downstream effector pathway, the nanoparticle delivery's potential impact, and the absence of human models, before it can be deemed suitable for publication in the EMBO Molecular Medicine.

We would like to thank the reviewer for summarizing our work and for valuable comments and recommendations on how to strengthen our manuscript.

Major concern:

1. Previous studies have investigated the angiogenic role of FOXF1 in other disease models or experimental models, as evidenced by publications from both the authors and other researchers (Bian et al., 2023, Nature Comms 14:2560; Sturtzel et al., Front. Bioeng. Biotechnol., 2018 Jun 14;6:76). It would be valuable for the authors to elucidate what factors or mechanisms contribute to the divergent roles of FOXF1 in different diseases, as this understanding could provide critical insights into its context-dependent functions and potential implications for therapeutic targeting in specific conditions.

We would like to thank the reviewer for the comment. We agree that this is a very important point. We have modified the Discussion section to incorporate published studies about the

divergent roles of FOXF1 in different diseases and potential implications for therapeutic targeting of FOXF1 in specific conditions as follows:

“In addition to its vessel-stabilizing property, FOXF1 was implicated in embryonic angiogenesis (Pradhan *et al.*, 2019; Ren *et al.*, 2014; Sturtzel *et al.*, 2018; Sun *et al.*, 2021), anti-fibrotic effects in lung fibroblasts (Black *et al.*, 2018), supporting lung regeneration after pneumonectomy (Bolte *et al.*, 2017; Cai *et al.*, 2016), and engraftment of endothelial progenitor cells after cell therapy (Kolesnichenko *et al.*, 2023; Wang *et al.*, 2021). Furthermore, FOXF1 was found to be required for oncogenic potential of tumor cells in rhabdomyosarcomas (Milewski *et al.*, 2017; Milewski *et al.*, 2021) and gastrointestinal stromal tumors (Ran *et al.*, 2018). These published studies demonstrate that FOXF1 plays diverse roles in different diseases by regulating multiple downstream signaling pathways in cell-specific manner reviewed in (Bolte *et al.*, 2020a) and (Bolte *et al.*, 2018). While therapeutic targeting of FOXF1 can be achieved using either gene therapy (Bolte *et al.*, 2020b) or TanFe small molecule compound (Pradhan *et al.*, 2023), further understanding of cell- and context-dependent functions of FOXF1 is needed for future considerations of FOXF1-directed therapies.”

We have also included the new references suggested by the Reviewer (Sturtzel *et al.*, *Front. Bioeng. Biotechnol.*, 2018 Jun 14;6:76 and Bian *et al.*, 2023, *Nature Comms* 14:2560) to support the revised Discussion.

*2. The downstream effector pathway of FOXF1 identified in this study, Wnt/ β -catenin signaling, is well-known for its significant role in regulating cancer cell growth and progression (Nusse *et al.*, *Cell* 2017 Jun 1;169(6):985-999). Moreover, there is existing evidence suggesting that FOXF1 can also regulate Wnt/ β -catenin signaling in other cell types (Shen *et al.*, *Ebiomedicine* 2020:102626). These findings raise concerns about the novelty of their current findings and the potential efficacy of solely targeting the endothelial-FOXF1-FZD4-Wnt/ β -catenin axis to effectively repress tumor growth. To provide a comprehensive analysis, the authors should address this crucial aspect in their discussion, considering the multifaceted roles of Wnt/ β -catenin signaling and FOXF1 in cancer biology.*

We agree. As the Reviewer suggested, we revised the Discussion section to provide additional information about the multifaceted roles of Wnt/ β -catenin signaling and FOXF1 in cancer biology as follows:

“In addition to its role in tumor vasculature, Wnt/ β catenin signaling regulates stromal and tumor cells to promote carcinogenesis (reviewed in (Nusse & Clevers, 2017)). Furthermore, FOXF1 was implicated in regulation of Wnt/ β catenin signaling in fibroblasts and osteoblasts (Reza *et al.*, 2023; Shen *et al.*, 2020). Therefore, it is possible that targeting of endothelial-FOXF1-FZD4-Wnt/ β catenin axis can be insufficient to effectively repress tumor growth. The use of multiple therapeutic agents simultaneously inhibiting tumor cells and cells of tumor microenvironment may be needed to supplement the endothelial-specific nanoparticle FZD4 therapy for NSCLC.”

We have also provided several new references (including Nusse *et al.*, *Cell* 2017 Jun 1;169(6):985-999) to support the revised Discussion.

3. The utilization of nanoparticles for delivering Fzd4 cDNA raises valid concerns, considering the

known toxicity associated with nanoparticles in human trials (p). Thus, the authors should address potential safety issues by investigating whether the administration of these nanoparticles leads to any pathological damage in the mice model used in the study. Additionally, given that the angiogenic role of Fzd4 is less characterized, it is crucial to assess whether these nanoparticles exert any adverse effects on normal blood vessel endothelial cells, particularly in the liver, as nanoparticles are known to accumulate in this organ.

We agree. In contrast to the previously characterized nanoparticles (metallic and liposome/micelle nanoparticles from the initial clinical trials) that raised safety concerns (Yang W., et al., *Annu Rev Pharmacol Toxicol.* 2021 Jan 6;61:269-289. doi: 10.1146, PMID: 32841092), the PBAE nanoparticles contains a large number of ester bonds that can be degraded by hydrolysis reactions under physiological conditions and turn into small molecules which are harmless or non-toxic to mammalian cells (Zhang et al., *Mol Ther Nucleic Acids.* 2023 Apr 23;32:568-581. doi: 10.1016, PMID: 37200860). Therefore, PBAE has been considered as a promising candidate for in vivo gene delivery with no or minimal toxicity (Young et al, *Adv Healthc Mater.* 2019 Jan;8(2):e1801359. doi: 10.1002, PMID: 30549448). In our previous work, no toxicity was found for the PBAE polymer-based nanoparticles (Deng Z., et al, *Bioact Mater.* 2023 Aug 7;31:1-17. doi: 10.1016; PMID: 37593494 and Donovan J et al, *Front Oncol.* 2023 Feb 2;13:1112859. PMID: 36816948).

Furthermore, we performed additional experiments to address the concern of the Reviewer about the potential nanoparticle toxicity. To test the safety of this intravenously injected PBAE nanoparticles carrying CMV- Fzd4 (nano-Fzd4) a toxicity study was performed to examine the liver and kidney metabolic panels in the peripheral blood (new Figure EV5, new Appendix Table S4, and new Appendix Table S5). Compared to the control group, no significant differences were found in total protein, Albumin, Globulins, ALP, Total Bilirubin, GGT, and ALT, indicating normal liver function. Similarly, no differences in BUN and Creatinine levels were found in the nano-Fzd4 treated group compared to the control group, indicating normal kidney function (new Appendix Figure EV5A, Appendix Table S4). Lack of abnormalities in these parameters indicated that the nano-Fzd4 treatment is not toxic to the liver and kidneys. In the hematology test, the nano-Fzd4 treated group had no changes in hematologic parameters compared to the control group (new Figure EV5B and Table S5). Furthermore, no significant body weight changes were found between nano-Fzd4-treated and control groups (new Figure EV5C). Also, the nano-Fzd4 treatment did not change the histological appearance of endothelial cells in the liver, as demonstrated by immunostaining for Pecam1 (CD31) (new Figure EV5D). These new data were incorporated into the revised manuscript.

4. The absence of human models and the use of a single mouse lung cell line in this study present significant challenges when attempting to translate the authors' findings into clinical applications. To address this limitation, the authors could enhance the relevance of their study by conducting additional experiments using human umbilical vein endothelial cells (HUVECs) or tumor-derived endothelial cells (TECs). Tube formation assays or in vivo matrix plug assays with these human cells would allow investigation of FOXF1's angiogenic role under conditions that better mimic the human microenvironment. Furthermore, to gain insights into the relevance of their findings in an in vivo setting closer to human disease, the authors can perform co-injection experiments of human cancer cells and genetically modified TECs in nude mice, allowing evaluation of the impact on subcutaneous lung tumor growth.

We agree and would like to thank the reviewer for the comment. As the reviewer suggested, we have performed additional experiments using human pulmonary arterial endothelial cells (HPAEC), which are more relevant to lung endothelial cells than HUVECs. The efficient shRNA-mediated inhibition of *FOXF1* decreased *FZD4* in HPAEC (new Figure 4F). Consistent with the reduced Wnt/ β -catenin signaling, expression of *Lef1*, one of Wnt/ β -catenin downstream targets, was also decreased (new Figure 4F). In addition, we have also performed the spheroid assay using H-441 human lung adenocarcinoma cells co-cultured with either control or *FOXF1*-deficient HPAEC. Tumor spheroids grew faster in the presence of *FOXF1*-deficient HPAEC (new Figure 4E), which was consistent with the conclusions from the animal model. Altogether, inhibition of *FOXF1* in human endothelial cells reduced *FZD4* and promoted tumor cell growth in human lung tumor spheroids. These new data have been incorporated into the revised manuscript.

As the reviewer suggested, we did perform the in vivo experiments co-injecting H-441 human cancer cells and genetically modified mouse TECs in immunocompromised mice. Unfortunately, due to small size of mouse lung tumors we were unable to purify sufficient numbers of TECs to adequately support human lung tumors in immunocompromised mice H-441 human cells. This could be due to substantial differences between subcutaneous and pulmonary microcirculation.

Minor concerns:

1. To enhance the clinical significance of their findings, the authors should incorporate survival data from their own patient cohort in addition to utilizing the online TCGA database (Figure 1E).

We would like to thank the Reviewer for the comment. Since we have been prospectively receiving the de-identified NSCLC lung tissue samples from the University of Cincinnati Biorepository, we do not have the survival data for the 18 samples that were included in the manuscript (Appendix Table S1). However, we have performed additional analysis for the 1,931 NSCLC samples from TCGA to assess separately the survival of patients with adenocarcinoma (AD) and squamous cell carcinoma (SCC). We have shown that by comparing patients in lower and upper quartile for *FOXF1* expression in both cohorts, including AD and SCC, the survival is significantly lower in the *FOXF1* low group (new Figure EV1A). These new data have been incorporated into the revised manuscript.

2. To strengthen their findings in Figure 2C, the authors should provide representative images of metastatic section stained with H&E.

As the Reviewer requested, we have provided the representative images of the H&E-stained metastatic sections (new Appendix Figure S1C).

3. The absence of scale bars in Figure 5A and B is a notable oversight that should be addressed. Furthermore, in Figure 5A and B, the presentation of a tumor adhering to the lung surface, rather than growing inside the lung, appears unusual and warrants clarification from the authors.

As requested, the scale bars in Figures 5A and 5B were added. Also, we have replaced the H&E-stained tumor section in Figure 5B to show the representative growth of lung tumor.

4. Once again, it is essential for the authors to include representative images of H&E-stained metastatic sections to substantiate their findings in Figure 5D.

As the Reviewer requested, we have provided the representative images of the H&E-stained metastatic sections (new Appendix Figure S1C).

5. To ensure transparency and reproducibility, the authors should include the "n" number, representing the total number of cells analyzed, for their single-cell sequencing results. This crucial information is currently missing in both the methods section and figure legend.

We agree and apologize for our oversight. The total number of cells that were used for single cell sequencing analyses had been added to the figure, the figure legend and to the methods section as the reviewer requested (revised Figure 1H).

6. The authors are advised to thoroughly review the manuscript for spelling and grammar errors. In particular, there is a spelling mistake in Figure 6E, where "survial" should be corrected to "survival."

The spelling mistake in Fig 6E has been corrected. The manuscript has been reviewed and the spelling and grammar errors were corrected.

Referee #2 (Remarks for Author):

Bian, Goda, Wang and colleagues report on the role of FOXF1 in lung endothelial cells in non-small cell lung cancer. Decreased expression of FOXF1 in tumor vessels altered vessel permeability and promoted cancer progression. Mechanistically, FOXF1 activated expression of Fzd4 and restoring its expression (using nanoparticle delivery of cDNA) was also sufficient to normalize tumor vessels. Overall, the manuscript is interesting, clearly written and describes important results.

Comments:

- *It would be interesting to know if the survival analysis presented in Fig 1D holds true in both adenocarcinoma and squamous cell carcinoma.*

We agree and would like to thank the reviewer for the valuable comment. As requested, we have performed additional analysis of NSCLC samples from TCGA to assess separately the survival of patients with adenocarcinoma (AD) and squamous cell carcinoma (SCC). We have shown that by comparing patients in lower and upper quartile for FOXF1 expression in both cohorts of AD and SCC, the survival is significantly lower in the FOXF1 low group (new Figure EV1A).

- *Why is SOX17 preferred to CD31 to identify endothelial cells in mouse tissues in Fig 1 but not later in Fig 3? Do these markers identify the same cells in mice?*

We agree and we have clarified this issue by providing new experimental data. Our goal was to quantify the number of FOXF1-positive endothelial cells. Since FOXF1 is a nuclear protein, we wanted to use the nuclear marker of lung endothelial cells for easier quantification of double positive cells. SOX17 is a nuclear protein and has been shown to be expressed in the majority of lung CD31-positive endothelial cells in mouse lungs (Lange et al, Dev Biol. 2014 Mar 1;387(1):109-20, PMID: 24418654; Cai et al, Sci Signal. 2016 Apr 19;9(424):ra40, PMID: 27095594). To verify our results and to address the reviewer's comment, we have performed additional staining using CD31 and FOXF1 antibodies. We have got the similar results and demonstrated that tumor-associated CD31-positive endothelial cells had decreased levels of FOXF1 protein compared to normal lung ECs (new Figure EV2A). These new data have been incorporated into the revised manuscript.

- *Fig 4C shows dual b-catenin/CD31 positivity, rather than nuclear b-catenin as is suggested in the results section text. Likewise, it is not clearly shown that nuclear b-catenin is reduced in the human data presented in Fig 4E-G. In Fig 5I and Fig 7F nuclear b-catenin is quantified but the same y-axis label is used as in previous figures that look at overall b-catenin expression. Ideally both quantifications would be shown for all experiments but this should be clarified.*

We agree. We apologize for the mistake in labeling the y-axes in Fig. 4C. The y-axis label should read “% of nuclear b-catenin⁺ endothelial cells”, which is the same as in Fig 5I and Fig 7F. Since b-catenin was expressed in both endothelial and tumor cells, we used dual b-catenin/CD31 staining to quantify the numbers of CD31-positive endothelial cells that expressed nuclear b-catenin. Also, we have added the higher magnification of tumor sections to demonstrate the presence of nuclear b-catenin staining in control endothelial cells and the decreased nuclear b-catenin staining in the endFoxf1^{+/-} endothelial cells (revised Figure 4B).

- *Did FOXF1 overexpression increase the expression of the other Wnt target genes identified by RNA seq analyses in Fig 4, or only Axin2 (Fig 5F)?*

We agree. To address this question, we have performed additional experiments and demonstrated that FOXF1 overexpression increased the expression Axin2 in lung endothelial cells (revised Figure 5F). Axin1, Lef1 and Wnt4 mRNAs were unchanged, indicating that this modest FOXF1 overexpression was insufficient to alter the expression of these genes. The new data was incorporated in the revised Figure 5F and the Results section.

- *The survival plots in Fig 1D and Fig 6E could be edited to appear similar (e.g. high expression in consistent color).*

We agree. As the Reviewer requested, we have changed the color of lines in Fig 1D and Fig 6E to make it consistent.

Typos:

- *"we next used [a] publicly available scRNAseq dataset"*
- *"Cre-mediated recombination of [a] floxed stop codon"*

- Fig 6E: "survi[v]al"
- "whether FOXF1 directly regulates transcription of [the] Fzd4 gene"
- "[A] previously published ChIP-seq dataset"

We apologize for our mistakes. The typos have been corrected.

Referee #3 (Comments on Novelty/Model System for Author):

Combination of patient data with orthotopic mouse model of NSCLC with lung EC-specific FOX-F1 overexpression or deficiency are powerful tools to directly address the research question. Use of nanoparticles for EC-specific induction of FOX-F1 expression may have a high medical impact on lung cancer therapy

Referee #3 (Remarks for Author):

This is a very interesting study addressing a role of FoxF1 activity in EC-TEC endothelial phenotypic transition observed in lung cancer. The authors demonstrate that FoxF1 deficiency is associated with higher mortality in patients with non-small cell lung cancers (NSCLC). They also observed similar findings in mice with EC-specific FOX-F1 deficiency, but tumor development was suppressed by FOX-F1 overexpression or gene transduction using EC-specific nanoparticles. The conclusions are supported by data, the rigor of study is strong, and impact high given the prospective of future use of EC-specific gene delivery to reduce mortality of patients with NSCLC. This reviewer has only a few minor comments to better clarify the mechanism and functional effects of EC-TEC endothelial phenotypic transition.

We appreciate the reviewer's affirmative comments and the time spent to review our manuscript.

Minor comments:

1. Was increased CD31 expression associated with increased EC proliferation? This scenario might be consistent with decreased capillary perfusion in FoxF1deficient lungs.

We agree. As the reviewer suggested, we performed additional experiment using immunostaining of control and FOXF1-deficient tumor sections for Ki-67 and CD31 to count the number of double positive cells. Proliferation rates of endothelial cells in lung tumors were low (less than 0.03%). Only single proliferating endothelial cells were present within FOXF1-deficient lung tumor sections and none in the control tumors. Thus, it is unlikely that the decreased capillary perfusion in the FOXF1-deficient tumors is a consequence of increased EC proliferation. We have added the new data to Appendix Figure S2 and have modified the Results section.

2. Given increased EC permeability in FoxF1deficient lungs, where there any changes in VE-cadherin and claudin-5 levels observed? If so, were they controlled by same FZD4 mechanism?

We agree and would like to thank the reviewer for the very insightful comment. To address the comment, we have performed additional analysis of human scRNAseq dataset and have shown that the expression levels of both VE-cadherin and Claudin-5 mRNAs were decreased in the tumor-associated ECs compared to normal lung ECs (new Appendix Figure S3A). Next, we verified that the mouse model recapitulated the human data by performing additional RNAscope experiments to examine VE-cadherin and Claudin-5 in FOXF1-deficient lung tumors. The new data are provided in the new Appendix Figure S3B-C. The data show that VE-cadherin and Claudin-5 levels are decreased in TECs of endFoxf1^{+/-} tumors compared to control tumors (new Appendix Figure S3B (left and middle panels) and Appendix Figure S3C). Finally, treatment with nanoparticles containing Fzd4 plasmid improved VE-cadherin and Claudin-5 levels (new Appendix Figure S3B (right panel) and Appendix Figure S3C), pointing to the same FRZ4 mechanism. These new data were incorporated into the Results section of the revised manuscript.

3. Besides EC-TEC transformation, did the authors observe increased angiogenesis in lung tumors? If so, how can this effect be reconciliated with inhibited capillary angiogenesis in FoxF1-deficient embryos?

We agree that this is an interesting question. We have observed elevated endothelial cell numbers in FoxF1-deficient tumors (Fig. 3), suggesting an increased tumor angiogenesis. This contrasts with diminished angiogenesis in FoxF1-deficient embryonic and neonatal lungs (Pradhan et al., 2019; Ren et al., 2014; Sturtzel et al., 2018; Sun et al., 2021). One explanation is that FOXF1 regulates angiogenesis differently in normal embryonic/neonatal lung tissue compared to lung tumor tissue. This is consistent with the different gene expression profiles in normal ECs (NEC) vs tumor-associated ECs (TEC) as reported in the present manuscript. Published studies demonstrate that FOXF1 can function as a pioneer transcription factor to remodel chromatin and make it accessible for other transcription factors to regulate gene expression (Ran et al, 2018). Therefore, it is possible that FOXF1-regulated chromatin remodeling can lead to either stimulation or inhibition of angiogenesis depending on other transcription factors that differentially expressed in normal and cancer lung tissues. We have incorporated this possibility into the Discussion section of the revised manuscript.

6th Mar 2024

Dear Tanya,

Thank you for submitting your revised manuscript. We have now received the feedback from the three referees who re-reviewed your manuscript. As you will see below, they are satisfied with the revisions, and I will therefore be able to accept your manuscript once the following editorial points will be addressed:

1/ Please address the minor comments from referee #2.

2/ Manuscript text:

- Your manuscript was cross-checked for text similarities, and a few matches were found with your previous Nat. Com. publication (see for instance screenshot attached). Please modify your text accordingly (please note that Materials and Methods are excluded from these checks).
- Please remove the red text, and only keep in track changes mode any new modification.
- Please provide up to 5 keywords.
- Materials and Methods:
 - o Mice: please indicate the origin of the animals and age at time of experiment (orthotopic model of lung cancer).
 - o Cells: please indicate origin, and whether the cells were authenticated and tested for mycoplasma contamination.
 - o Statistics: please include a statement on inclusion/exclusion criteria.
 - o Human samples: please include a statement that the experiments conformed to the principles set out in the WMA Declaration of Helsinki and the Department of Health Services Belmont Report.
- Data availability: Thank you for providing a reviewer token for your RNAseq data. Please note that this dataset must be made public before acceptance of the manuscript.
- Please replace "Conflict of Interest" by "Disclosure statement and competing interests". We updated our journal's competing interests policy and request authors to consider both actual and perceived competing interests (<https://www.embopress.org/competing-interests>). Please add the sentence: "T. Kalin is a member of the EMM Editorial Board. This has no bearing on the editorial consideration of this article for publication."

3/ Figures and Appendix:

- Please provide exact p values, not a range, in the figures or in their legends.
- Appendix: please add page numbers to the table of content, remove the red font, and upload in pdf format.
- Please make sure the figures are referenced in chronological order in the text (currently, Fig EV2A is called out before Fig EV1B,C).
- Reuse of figures/figure panels is allowed, but should be mentioned in the figure legends (i.e. Figure 6C Mouse lung and Figure EV2 Normal lung).
- Please carefully check the composition of your Appendix Figure S2.
- Please address the queries from our data editors in the figure legends:
 1. Please note that a separate 'Data Information' section is required in the legends of figures 3d-g; EV 3d-e.
 2. Please indicate the statistical test used for data analysis in the legend of figure 4a.
 3. Please note that the box plots need to be defined in terms of minima, maxima, centre, bounds of box and whiskers, and percentile in the legends of figures 1h; 6a; EV 1c.
 4. Please note that information related to n is missing in the legends of figure 6a; EV 1c.
 5. Please note that the scale bar needs to be defined for figures 4b, g; 5a-b, e; 6c-d; 7c, e-g; EV 2f.
 6. Please note that scale bar and its definition are missing for figure 2h.
 7. Please note that the white arrows are not defined in the legend of figure 7f, EV 5d. This needs to be rectified.

4/ Thank you for providing Source Data. Please upload them as one file per figure.

5/ Checklist:

- please provide information on cell authentication/mycoplasma contamination.
- please check the section "experimental animals/animal observed in or captured from the field", as I don't think it applies to your manuscript.
- please provide information in the "statistics" section, inclusion/exclusion criteria.
- please check the section "Ethics/Specimen and field samples", as I don't think it applies to your study.

6/ Thank you for providing a synopsis. Please also suggest a visual abstract to illustrate your article as a PNG/tiff/jpeg file 550 px wide x 300-600 px high.

7/ As part of the EMBO Publications transparent editorial process initiative (see our Editorial at

<http://embomolmed.embopress.org/content/2/9/329>), EMBO Molecular Medicine will publish online a Review Process File (RPF) to accompany accepted manuscripts.

This file will be published in conjunction with your paper and will include the anonymous referee reports, your point-by-point response and all pertinent correspondence relating to the manuscript. Let us know whether you agree with the publication of the RPF.

I look forward to receiving your revised manuscript.

With kind regards,

Lise

***** Reviewer's comments *****

Referee #1 (Comments on Novelty/Model System for Author):

This study holds significance for the medical application of the blood vessel normalization concept. However, the utilization of nanoparticles may still pose certain limitations in clinical settings due to their known toxicity. As I mentioned before, the downstream effector pathway of FOXF1 identified in this study, Wnt/ β -catenin signaling, is well-known for its significant role in regulating cancer cell growth and progression (Nusse et al., Cell 2017 Jun 1;169(6):985-999). Moreover, there is existing evidence suggesting that FOXF1 can also regulate Wnt/ β -catenin signaling in other cell types (Shen et al., Ebiomedicine 2020:102626). These findings raise concerns about the novelty of their current findings and the potential efficacy of solely targeting the endothelial-FOXF1-FZD4-Wnt/ β -catenin axis to effectively repress tumor growth. I have no objections regarding the choice of model organism used in this study.

Referee #1 (Remarks for Author):

The authors have satisfactorily addressed my concerns in this revised version.

Referee #2 (Remarks for Author):

The authors have addressed my concerns and those of other reviewers well and I now feel the paper is suitable for publication.

Some minor thoughts from my read through of this version:

- * There is a typo in Figure 1E ("suivivor").
- * The inconsistent use of bold/not bold font in Figures is quite distracting.
- * Define BUN

The authors addressed the minor editorial issues.

21st Mar 2024

Dear Tanya,

Thank you for submitting your revised files. I am pleased to inform you that your manuscript is accepted for publication and is now being sent to our publisher to be included in the next available issue of EMBO Molecular Medicine!

Thank you for resizing the synopsis, however, the text appears very small (difficult to read), and the resolution is low. Could you please send us a new version as soon as possible?

Please also note that I have removed "Source data are available online for this figure" from the figure legends.

With kind regards,

Lise
